# Contrasting life-history responses to climate variability in eastern and western North Pacific sardine populations

Tatsuya Sakamoto [1] ✉, Motomitsu Takahashi [1], Ming-Tsung Chung [2,3], Ryan R. Rykaczewski [4,5], Kosei Komatsu[6,2], Kotaro Shirai [2], Toyoho Ishimura [7,8] & Tomihiko Higuchi[2]

Massive populations of sardines inhabit both the western and eastern boundaries of the world's subtropical ocean basins, supporting both commercial fisheries and populations of marine predators. Sardine populations in western and eastern boundary current systems have responded oppositely to decadal scale anomalies in ocean temperature, but the mechanism for differing variability has remained unclear. Here, based on otolith microstructure and high-resolution stable isotope analyses, we show that habitat temperature, early life growth rates, energy expenditure, metabolically optimal temperature, and, most importantly, the relationship between growth rate and temperature are remarkably different between the two subpopulations in the western and eastern North Pacific. Varying metabolic responses to environmental changes partly explain the contrasting growth responses. Consistent differences in the life-history traits are observed between subpopulations in the western and eastern boundary current systems around South Africa. These growth and survival characteristics can facilitate the contrasting responses of sardine populations to climate change.

Populations of marine organisms are dynamic, and the causes of their variability are often difficult to understand. Sardines (*Sardinops, Sardina* spp.) are small, planktivorous pelagic fish known for intense population fluctuations with considerable economic[1] and ecological[2] impacts. They are globally distributed in all temperate oceans except the western Atlantic with shallow genetic divergence[3], including regions of distinct oceanographic conditions, i.e., the western boundaries dominated by a warm current and the eastern boundaries characterised by a cool current with associated coastal upwelling (e.g., Fig. 1a, b). Stock assessments in recent decades[4,5], time series of fishery catches in the 20th century[6], and palaeological records in seabed sediment cores[7–10] all suggest that abundances of sardine populations worldwide have fluctuated by several orders of magnitude at multidecadal scales. As the boom-and-bust cycles have repeatedly occurred over several thousand years[7,8] and exhibited correlations with basin-scale climate indices such as the Pacific Decadal Oscillation (PDO) (e.g., Chavez et al.[11]), they have been considered natural phenomena mainly driven by environmental variability. Nevertheless, the mechanisms connecting physical forcing with sardine populations remain unclear.

The sardine populations in western and eastern boundary current systems are known to present a marked difference in their response to climate change[11]. The two sardine subpopulations in the western and eastern North Pacific, namely the Pacific subpopulation of Japanese

[1]Japan Fisheries Research and Education Agency, Nagasaki, Japan. [2]Atmosphere and Ocean Research Institute, The University of Tokyo, Chiba, Japan. [3]Institute of Oceanography, National Taiwan University, Taipei, Taiwan. [4]Pacific Islands Fisheries Science Center, NOAA National Marine Fisheries Service, Honolulu, HI, USA. [5]Department of Oceanography, School of Ocean and Earth Science and Technology, University of Hawaii, Honolulu, HI, USA. [6]Graduate School of Frontier Sciences, University of Tokyo, Chiba, Japan. [7]Graduate School of Human and Environmental Studies, Kyoto University, Kyoto, Japan. [8]Department of Chemistry and Material Engineering, National Institute of Technology, Ibaraki College, Ibaraki, Japan. ✉e-mail: tatsfish@gmail.com

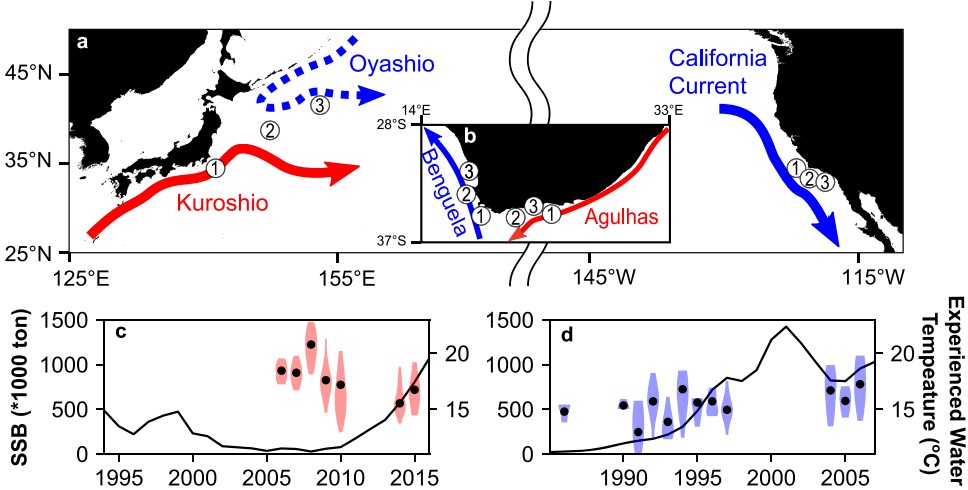

**Fig. 1 | Schematics of the boundary currents near the habitats of sardine populations and the biomass fluctuations during the study period.** Schematics of boundary currents nearby habitat areas of JP sardine (the Pacific subpopulation of Japanese sardine) and CA sardine (the northern subpopulation of Pacific sardine) (**a**) and South African sardines (western and south-eastern subpopulations) (**b**). The numbers show the rough location of sardine in 1: larval stage (Standard Length (SL) < 35 mm), 2: early juvenile stage (SL: 35–60 mm), and 3: late juvenile stage (SL: 60–85 mm). In **a**, **b**, western boundary currents are shown as solid red arrows and eastern boundary currents as solid blue arrows. (c, d) Time series of spawning stock biomass (SSB, solid lines) of JP sardine[4] (**c**) and CA sardine[5] (**d**) together with violin plots of mean experienced water temperature from hatch to 120 days post hatch for JP sardine (**c**) and from hatch to 150 days post hatch for CA sardine (**d**) estimated based on otolith $\delta^{18}O$. Shades and filled plots in **c**, **d** represent the density distributions of individual data and year-class means.

sardine *Sardinpos sagax melanostictus* (hereafter JP sardine), and the northern subpopulation of Pacific sardine *Sardinops sagax sagax*, (hereafter CA sardine), have attracted great attention due to their commercial importance and multidecadal biomass fluctuations. JP sardine spawn in winter to spring in the inshore area of the Kuroshio. Their offspring are transported eastwards by the Kuroshio, and the individuals then migrate northwards to the subarctic Oyashio region (Fig. 1a). CA sardine mainly spawn in spring to summer off the southern California coast, and their offspring are considered to move inshore (Fig. 1a). Based on variable statistical methods, a number of studies have found significant linear or non-linear links between habitat temperatures and sardine biomass, recruitment or early life survival[11–17]. However, it has long been an enigma why JP sardine biomass increases during cooler periods[12,13,17], while biomass of CA and other sardine populations in eastern boundary current systems increase during warmer periods[11,15,16]. Examining this puzzling phenomenon by comparing the eastern and western boundary current populations provides a unique opportunity to understand how marine fish respond to environmental variability. Such an understanding is necessary to better project population responses to future climate change.

Each mature female sardine is estimated to spawn hundreds of thousands (and potentially more than a million) of eggs each year[18–21]. The vast majority of these newly spawned individuals die during the larval and juvenile stages, while a very small proportion survive and recruit. Variability in the recruitment is one of the major factors regulating fish populations, along with density-dependent mortality of the recruits[22]. Studies have shown that only individuals that had an exceptionally high growth rate during the larval and juvenile stages survived the mass mortality period[23]. Faster growth is hypothesised to confer higher survival rates through indirect advantages associated with larger body size[24] and shorter duration of vulnerable life stages[23] or by becoming resilient to predation[25]. As the early life growth rate has been shown to be strongly correlated with recruitment of JP sardine[26] especially during periods of low biomass[27], the effects of environmental variability on early life growth are hypothesised to be critical factors regulating early life survival and recruitment, and potentially affecting population-level dynamics.

The growth variations with environmental conditions depend on the energy acquirement and expense. Energy use of marine ectotherms is constrained by the aerobic scope, the capacity to increase metabolic rate beyond the maintenance level as defined by the difference between standard and maximum metabolic rates[28]. Aerobic scope is maximised at a specific temperature (i.e., the optimal temperature) and decreases below or above it[28], but the pattern varies with life stages[29]. Therefore, temperature variation can affect fish growth directly through metabolism[29] or indirectly via food availability. A recent study of sardine *Sardinpos sagax ocellatus* off South Africa, which includes both a western and an eastern boundary current system (Fig. 1b), showed that the sardine subpopulation off the south and east coasts (the warm, western boundary current system) grew faster under approximately 3 °C warmer water than that off the west coast (the cool, eastern boundary current system)[30]. As the differences follow the distinct oceanographic conditions of the western and eastern boundary current systems, they may be the common features of populations in subtropical boundary current systems around the globe. In systems of such dissimilar nursery environments, seawater temperature variability can impact fish physiology and food availability in distinct ways, potentially leading to different responses of growth. However, it has been challenging to explore this concept due to the critical lack of techniques to track the environment experienced by fish during early life stages and their physiological condition.

The application of novel analyses to otoliths, the calcium carbonate structures in the inner ear of fish, allows a breakthrough on this research front. Sardine otoliths form growth increments on a daily basis during larval and juvenile stages[31–33]. As body length and otolith radius show strong correlations for sardines <19 cm standard length (SL)[26,31], their somatic growth trajectories can be inferred from daily increment widths of otoliths[34]. Furthermore, ambient water temperature and field metabolic rate, which is the energy expenditure of free-ranging animals in their natural environment[35], can be estimated from the oxygen[36,37] and carbon[38] stable isotope ratios ($\delta^{18}O$, $\delta^{13}C$) of otoliths, respectively. Recent technical developments of such isotope analyses and microscale sampling have dramatically improved the resolution such that variability at age periods as small as 10 days can be examined[39], thereby opening the door for the reconstruction of thermal and metabolic histories during the larval and juvenile stages.

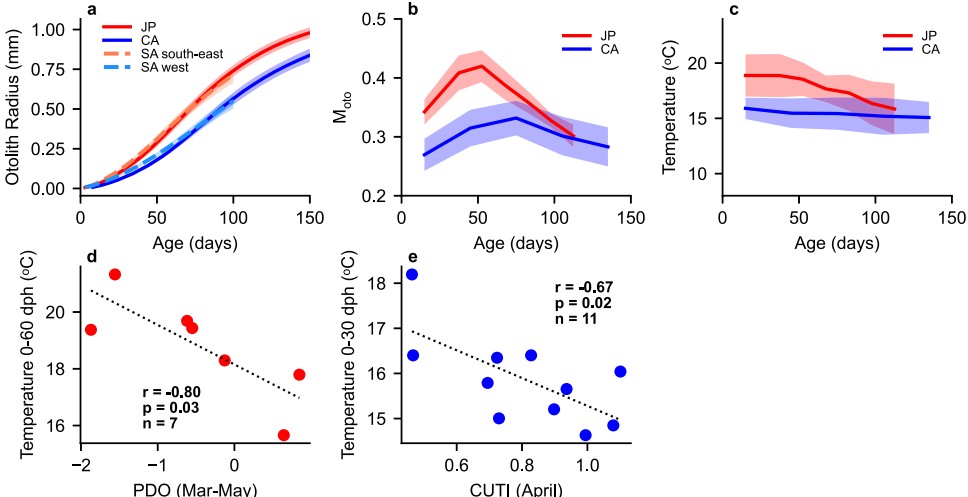

**Fig. 2 | Life history traits of sardine populations and relationships between experienced temperature and evironmental indices. a** Mean otolith growth trajectories of JP, CA, and South African sardines. **b** Mean estimated metabolic proxy $M_{oto}$ of JP and CA sardines. **c** Mean water temperature experienced by JP and CA sardines. The shades show 1 standard deviation ranges (**a–c**). **d** Relationship between year-class mean experienced water temperature of JP during 0-60 dph and Pacific Decadal Oscillation index[41] during March to May. **e** Relationship between year-class mean experienced water temperature of CA during 0–30 dph and Coastal Upwelling Transport Index[42] at 33–36 °N in April. Pearson's r values are shown with p-values (two-sided) without adjustments for multiple comparisons, and *n* is the number of year-classes (**d**, **e**).

Here, we investigated the early life history traits and their responses to seawater temperature variation of sardines in western and eastern boundary current systems to better understand the factors influencing the populations' apparent opposite responses to decadal scale anomalies in ocean temperature. Our main focus was on comparison of JP and CA sardines in the western and eastern North Pacific. Otoliths were collected from 420 age-0 recruited JP sardine (SL: 10–15 cm) sampled in the Oyashio region during 2006–2010, 2014 and 2015 (Supplementary Fig. 1; Supplementary Table 1) and from 136 recruited age-1 CA sardine (SL: 10–16 cm) captured in the coastal water of the Southern California Bight during 1987, 1991–1998 and 2005–2007 (Supplementary Table 2). Note that the body sizes and life history stages of samples were comparable between regions, and sampling years for both populations included the periods of biomass increases[4,5] (Fig. 1c, d) to cover year-classes that likely experienced favourable conditions for population-level growth and survival. Daily increment widths and $\delta^{13}C$ and $\delta^{18}O$ were measured for these samples from their hatching until 120–150 days post hatch (dph). This period includes the entire larval and most of the juvenile stages that likely cover the mass mortality period during which variations in mortality rates play a great role in determination of year-class strength of fish[40]. Measurements of otolith $\delta^{13}C$ and previously published data of $\delta^{13}C$ of sardine prey and dissolved inorganic carbon (Supplementary Fig. 2) allowed estimation of the ontogenetic change of $M_{oto}$ (see Methods), which is the proportion of metabolically derived carbon in otolith carbonate and represents the proxy of field metabolic rate[38]. Measurements of $\delta^{18}O$ of otoliths, together with measurements of seawater $\delta^{18}O$ (Supplementary Fig. 3) and observations of temperature and salinity by Argo floats (Supplementary Fig. 4), permitted reconstruction of the history of the water temperature of each individual's habitat (see Methods, Supplementary Table 3). This suite of measurements showed that habitat temperature, early life growth rates, energy expenditure, metabolically optimal temperature, and most importantly the relationship between the growth rate and temperature were remarkably different between JP and CA sardines. Some of these life-history traits were compared to those of subpopulations off the south-east and the west coasts off South Africa to infer general features of sardine populations in the western and eastern boundary current systems.

## Results and Discussion
### Life-history traits of sardines in the western and eastern boundary current systems
First, we tested for differences in basic early life growth, $M_{oto}$ and nursery temperatures between JP and CA sardines. Mean otolith radii were higher in JP sardine throughout 10–150 dph (Fig. 2a). Despite the vast geographic distances that separate sardine populations in South Africa from those in the North Pacific, growth histories of otolith radii of JP sardine and sardine from the south-east coast of South Africa[30] were similar until 100 dph. Likewise, the growth histories of CA sardine and sardine from the west coast of South Africa[30] were similar until 100 dph (Fig. 2a). These characteristics were evident by the significant differences of otolith increment widths among regions, and the similarity between JP sardine and sardine from the south-east coast of South Africa and between CA sardine and sardine from the west coast of South Africa especially during 21–60 dph (Supplementary Figs. 5, 6; Supplementary Table 4). Mean $M_{oto}$ peaked at 45–60 dph in JP sardine and 60–90 dph in CA sardine, both around the early juvenile stage, and were significantly higher in JP sardine during 0–120 dph (Fig. 2b, Supplementary Fig. 7, Supplementary Table 5). Mean experienced water temperature of JP sardine gradually declined from around 19 °C to 16 °C in the first 120 days, likely reflecting the known northward movement towards the Oyashio region. In contrast, those of CA sardine were relatively constant at about 15–16 °C in the first 150 days, suggesting the residency around frontal areas of coastal upwelling water (Fig. 2c). The temperature was significantly higher in JP than in CA sardine during 0–90 dph (Supplementary Fig. 8, Supplementary Table 6). The differences in temperature and $M_{oto}$ histories and otolith growth trajectories suggest JP sardine has a warmer nursery environment, higher metabolic rate, and faster growth rate than CA sardine. These are consistent with the differences between populations of the western and eastern boundary current systems that support sardine near South Africa[30], likely reflecting the differing ocean environments in the western (warmer) and eastern (cooler) boundary current systems.

One of the primary metrics frequently used to characterise interannual variability in the North Pacific is the PDO. The PDO positive phase is generally associated with cool SST anomalies in the western North Pacific and anomalously warm SST in the eastern portion of the

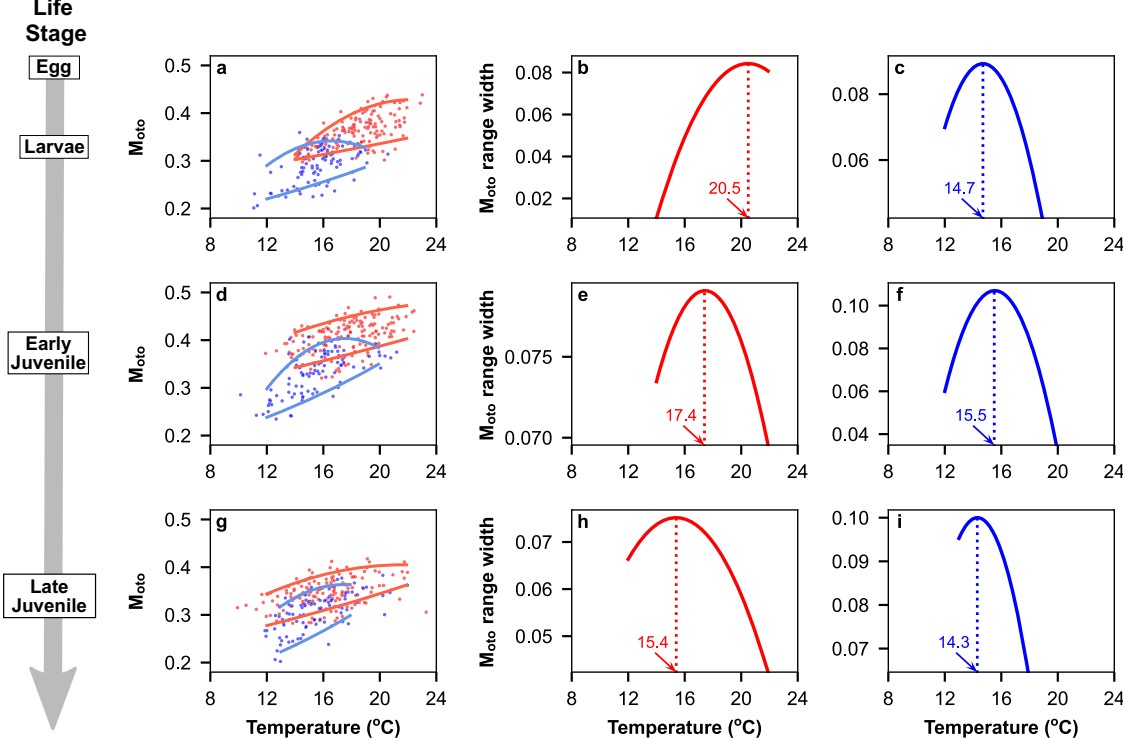

**Fig. 3 | Estimation of optimal temperatures based on metabolic proxy M$_{oto}$ and experienced temperature.** The relationships of M$_{oto}$ and experienced temperature for JP (red) and CA (blue) sardines during larval (**a**), early juvenile (**d**), and late juvenile (**g**) stages. The upper lines are the polynomial regression for the 95th percentile M$_{oto}$ grouped at each 1 °C bin, while the lower lines are the regression for the 5th percentile values of each bin calculated by a generalised linear model with Gaussian distribution and a log link. The dashed lines are the linear regression lines for overlapping temperature ranges (dark red: JP, dark blue: CA). The difference between the regression lines for 95th and 5th percentile values of each bin for JP sardine during larval (**b**), early juvenile (**e**), and late juvenile (**h**) stages and for CA sardine during larval (**c**), early juvenile (**f**), and late juvenile (**i**) stages. The dotted line shows temperature at the peak of the range width.

basin[41]. Temperatures experienced by JP sardine during 0–60 dph were negatively correlated with PDO from March to May (Fig. 2d, Pearson's r = −0.80, p = 0.03, n = 7). While temperatures experienced by CA sardine during 0–30 dph showed no significant correlation with PDO during April, they were negatively correlated with the Coastal Upwelling Transport Index[42] off Southern California during April (Fig. 2e, Pearson's r = −0.67, p = 0.02, n = 11). As SST in the entire western North Pacific is cool during PDO positive phase[41], and the California Current region becomes cooler when coastal upwelling is stronger, these significant relationships are reasonable and substantiate that otoliths provide a record of the physical conditions experienced during development.

**Life-history responses to temperature variations**

Next, to understand the relationship between temperature and fish metabolism, we compared M$_{oto}$ with the temperature fish experienced during their early life stages. Generally, M$_{oto}$ became higher with the increase of temperature (Fig. 3a, d, g), suggesting higher metabolic rates at warmer temperatures. JP sardine in the early and late juvenile stages had significantly higher M$_{oto}$ than CA sardine even after taking the effect of temperature into account (Fig. 3a, d, g, generalised linear model, Supplementary Fig. 9, Supplementary Tables 7, 8), showing that JP sardine generally had greater energy expenditure. Here, the lowest field metabolic rate would be close to the standard metabolic rate and the highest field metabolic rate would be constrained by maximum metabolic rate[42]. The gap between the lowest and highest M$_{oto}$ values, therefore, provides an analogue of the aerobic scope and would be maximised at the optimal temperature[43]. Thus, the optimal temperature for JP sardine was found at >20 °C during the larval stage but gradually declined to approximately 15 °C with age (Fig. 3b, e, h, Supplementary Fig 10),

while that of CA sardine remained around 14–15 °C (Fig. 3c, f, i, Supplementary Fig. 10). This difference reveals that the two sardine populations have different thermal preferences and are likely to have adapted to different habitat temperatures.

Finally, we tested the relationship between temperature and growth during early life stages. Back-calculated length and mean experienced temperature from hatch to about the end of the larval stage (JP: 45 dph, CA: 60 dph) showed positive correlations in both JP (Pearson's r = 0.46, corrected-p value (p$_c$) = 0.045, n = 26, Supplementary Table 9) and CA sardines (Pearson's r = 0.42, p$_c$ = 0.046, n = 21) (Fig. 4a, b), suggesting that warmer temperatures enhance larval growth in both regions. However, the relationships between length and mean temperature in JP sardine changed with later life stages, becoming dome-shaped around the end of the early juvenile stage (75 dph; linear model with quadratic term, p$_c$ = 0.046, Supplementary Table 10) and showed a significant negative correlation at the end of the late juvenile stage (105 dph; Pearson's r = −0.58, p$_c$ = 0.011, n = 26). In contrast, the relationship for CA sardine showed a significant positive correlation at the end of the early juvenile stage (90 dph; Pearson's r = 0.47, p$_c$ = 0.045, n = 21) and late juvenile stage (120 dph; Pearson's r = 0.43, p$_c$ = 0.046, n = 21). Length is the integration of daily somatic growth. These results, therefore, reveal that JP and CA sardines both had higher growth rates during the larval stage at warmer temperature (Fig. 4a, b), although their mean growth rate throughout larval, early and late juvenile stages was faster under cooler conditions in JP sardine but under warmer conditions in CA sardine (Fig. 4e, f). For each life stage, the temperatures of both highest median somatic growth rate and highest median otolith growth rate of JP sardine declined from >20 °C during the larval stage to around 15 °C during late juvenile stages, closely following the shift of optimal temperature (Supplementary Figs. 11a, c, e and 12a, c, e), while those of CA sardine were

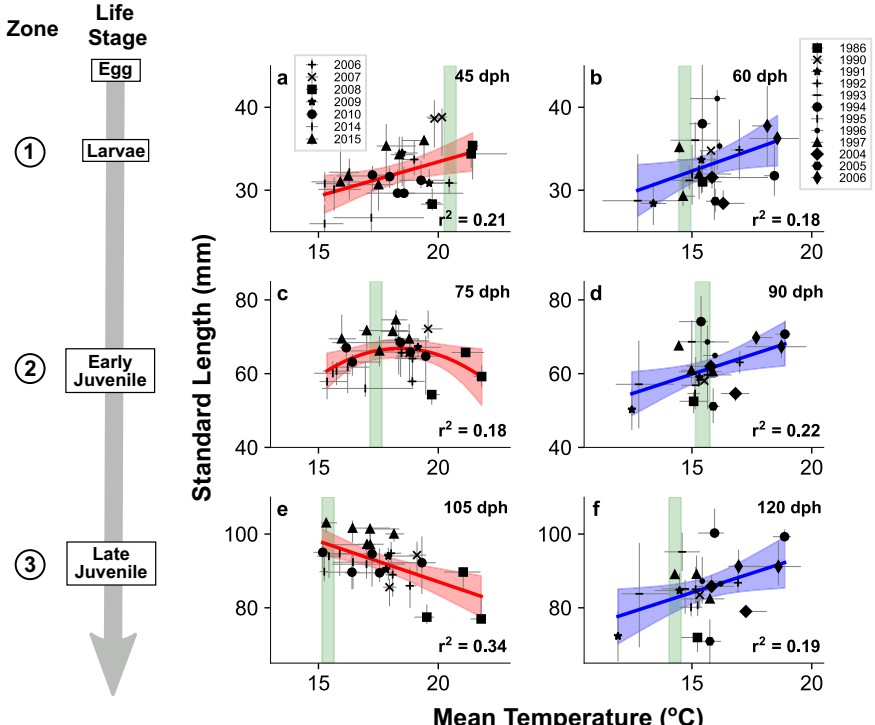

**Fig. 4 | Responses of sardine size to temperature variability.** Comparisons between standard length and mean experienced water temperature from hatch through late larval stage of JP (**a**) and CA (**b**) sardines, from hatch through early juvenile stage of JP (**c**) and CA (**d**) sardines, and from hatch through late juvenile stage of JP (**e**) and CA (**f**) sardines. Plots and error bars represent median and 25th and 75th percentiles in each sampling batch. Significant regression lines (*p* < 0.05, linear or quadratic regression analysis, two-sided tests and corrected for multiple comparisons using the Benjamini-Hochberg procedure with α = 0.05) and 95% confidence intervals are shown as solid lines and shades. Zone numbers are the oceanographic zones in Fig. 1a where sardine in each stage were typically distributed. The light green bars indicate the optimal temperatures predicted for each stage in Fig. 3b, c, e, f, h, i.

higher than the optimal temperature in larval and early juvenile stages (Supplementary Figs. 11b, d, f and 12b, d, f).

Thus, we found differences in the responses of early life growth to ambient temperature variations between JP and CA sardines, and these differential responses depended on life stage. Although events in the larval stage have long been assumed to be responsible for recruitment fluctuations[44], year-class strength cannot be predicted from the abundance of larvae in CA and JP sardines[31,45]. Studies also indicate that growth rates of early juveniles are positively correlated with recruitment of JP sardine[26], especially during low biomass periods[27]. These results suggest that high growth rate throughout larval and juvenile stages, not only during the larval stage, is important to improve recruitment success. Furthermore, sea surface temperatures in the main distribution areas of JP and CA sardines during 4 months from the main spawning month (see Methods), showed significant negative and positive correlations with early life survival index (log recruitment residuals (LNRR)) of JP and CA sardine, respectively (JP (Mar–Jun): Pearson's r = −0.45, *p* = 0.007, *n* = 35, CA (Apr–Jul): Pearson's r = 0.49, *p* = 0.008, *n* = 28, Supplementary Fig. 13). This suggests that cooler (for JP sardine) and warmer (for CA sardine) habitat temperatures are linked to improvements of early survival and recruitment. Based on these observations, we conclude that because conditions promoting fast growth through critical life stages are associated with differing environmental temperatures, survival of the early life history stages of JP and CA sardine increase in cooler and warmer regimes, respectively. Such changes in survival rates are likely critical factors influencing the differing responses of recruitment in these populations to temperature changes, and potentially those of overall biomass. The similarity of life history traits of JP and CA sardines and of subpopulations off western and south-eastern coasts of South Africa

suggests that the mechanism may universally function in western and eastern boundary current systems worldwide.

Responses of wild fish's growth to temperature variations are variable among species and ecosystems[46]. In the case of JP and CA sardines, the interaction of fish metabolism and food availability may have formed the contrasting responses of growth to temperature. For JP sardine, the ontogenetic change in the relationship between temperature and growth closely matched the decline of optimal temperature (Fig. 4a, c, e; Supplementary Fig. 11a, c, e). This suggests JP sardine often fully uses the aerobic scope especially around its optimal temperatures, and thus its total energy expenditure is constrained by the width of aerobic scope. This is likely due to the high-level foraging activities, digestion and growth under rich energy resources available during the transportation and migration from the subtropical Kuroshio to the subarctic Oyashio. Sardine mainly prey on small zooplankton[47], and the frontal area of the Kuroshio with waters warmer than 18 °C provides abundant small copepods that are suitable for JP sardine larvae[48]. The main prey species of JP sardine larvae and juveniles are especially abundant at 15–17 °C in the Kuroshio-Oyashio transition zone[49], which coincides with the optimal temperatures during the juvenile stage (Fig. 3e, h). On the other hand, larger size and faster growth CA sardine was consistently found at temperatures above the optimal temperature (Fig. 4b, d, f; Supplementary Fig. 11b, d, f), and the energy expenditures of CA sardine were smaller than those of JP sardine (Fig. 3 a, d, g). These findings suggest that CA sardine generally does not fully use the aerobic scope and thus the width of the scope is not the primary constraint of the energy expenditure. Previous studies indicate that sardine distribution and energy acquirement may be limited in cooler coastal upwelling waters in the eastern boundary current systems due to the large plankton sizes that stimulate a mode of inefficient particulate

feeding[47] or low oxygen concentration[50]. Warmer habitats offshore that are dominated by smaller plankton, to which nutrient supply has been shown to be related to sardine production[51], may offer less stressful conditions. Thus, we consider that growth rates during the early life stage of JP sardine increase during cooler regimes because the wider aerobic scope of juveniles and the expansion of prey-rich water allow greater energy use for growth, while growth rates of CA sardine increase during warmer regimes because the aerobic scope, which is generally used to a limited extent, is used more under the less stressful habitat conditions. The possibility remains that the use of literature values to estimate $M_{oto}$ affected the estimation of optimal temperatures for CA sardine. However, for both JP and CA sardine, error ranges in $M_{oto}$ estimations were limited (see Methods), $M_{oto}$ increased appropriately with temperature with varying diversity across the temperature range, and the optimal temperatures determined matched well with the temperatures of their distribution along ontogenetic stages. These suggest the results are robust and that there are mechanical differences in physiological response between the sardines. Temperature sets the capacity of energy expenditure, actual utilisation depends on prey availability, and it may be the combined effect of these factors that ultimately determines growth.

## Implications for the response of sardine populations to climate change

While there remains low scientific consensus regarding dynamics of these fisheries in future, our results offer a few clues for the fate of sardine populations under climate change. The western boundary currents have warmed faster than the global mean over the last century[52], and the mixed layer depth in winter is predicted to decrease in the Kuroshio-Oyashio transition zone[52]. These changes may limit the nutrient supply necessary for the plankton productions during spring to summer[53]. In contrast, rates of warming in the coastal areas of the eastern boundary upwelling systems have been slower[52], and several models project increases in nutrient supply and plankton productivity in the California Current system under global warming conditions[54]. Coupling these findings with our conclusion regarding metabolic responses to changing temperature indicate that growth of JP and CA sardine may tend to decrease and increase under global warming, respectively. Such changes in growth rates may impact the survival of early life history stages and recruitment, and potentially have consequences for population-level dynamics. During the past 30–40 years, large variations in the early life survival of the sardines were observed within the SST anomalies within ± 1.5 °C in each habitat (Supplementary Fig. 13). As the rates of increase in SST in the regions are predicted to be 0.03–0.04 °C/year in the 21st century under the most severe $CO_2$ emission scenario (SSP5-8.5)[52], the impacts of the long-term temperature trend on sardine populations could become stronger than those of the decadal variability in 30–50 years. Changes in growth and survival during later life history stages, phenology, spatial distribution, maternal condition, and fishing pressure must be considered when considering future population changes. However, the survival of *Sardinops* genus is unlikely to be threatened by future climate changes. Our results suggest that JP and CA sardines, and perhaps other populations in the western and eastern boundary current systems such as South African sardine off the south-eastern and western coasts, are likely to show contrasting responses. Furthermore, there are other sardine subpopulations in many systems that inhabit warmer waters and could potentially flourish under a warmer climate (e.g., the southern subpopulation of Pacific sardine). The ability to adapt to diverse thermal environments results in differing responses to environmental variability among populations and confer robustness to climate changes to the genus, which could be the key element of the evolutionary strategy of sardines.

Population responses to changing ocean temperatures have differed, even for closely related species of the same genus. Despite this somewhat confusing observation, resolving the sensitivity of populations to climate change and variability is possible with an improved understanding of ecosystem dynamics and the responses of the organisms' metabolic rates. In the case of the sardine populations of the western and eastern North Pacific, understanding the differing biological conditions associated with changing oceanographic processes provided insight into a long-standing challenge faced by biological oceanographers. Overall, our analyses for sardines in the North Pacific and off the South Africa coasts revealed differences in habitat temperature, early life growth rates, metabolically optimal temperature, and the response of growth to temperature variation between sardines in the western and eastern boundary current systems, shedding light on the potential mechanism that explains why these populations increase during opposite temperature regimes. Our findings point to the importance of considering interactions between processes addressed by two major theories for population fluctuation of small pelagic fish that seemed to be conflicting, the optimal temperature hypothesis[55] and the trophodynamics hypothesis[47]. Although understanding the dynamics of the underwater world is still exceedingly difficult, technical developments will allow stepwise improvements necessary to resolve the impacts of climate change and variability on marine populations and support prudent and more sustainable relationships between that natural world and human society.

## Methods
All procedures accorded to administrative provision of animal welfare of the Fisheries Research Education Agency Japan. All statistical tests used in this study are two-sided.

### Otolith samples
From the western North Pacific, age-0 JP sardine were collected from samples taken during acoustic and sub-surface trawl surveys in the offshore Oyashio region conducted during 2006–2010 and 2014–2015. The surveys were conducted by Japan Fisheries Research and Education Agency every autumn since 2005 which aim to estimate the abundance of small pelagic species. The abundance of young-of-the-year sardine in the region in the season, approximately 10–15 cm in standard length (SL), is considered a proxy for the abundance of recruits of the Pacific stock and used to tune the cohort analysis in stock assessment[4]. As representatives of the young-of-the-year population in the region, 2–6 trawl stations each year that had relatively larger catch-per-unit-effort were selected (Supplementary Fig. 1), and 9–20 individuals were randomly selected from each station for otolith analyses (Supplementary Table 1). Age of fish was initially judged by SL (10–15 cm) and later confirmed by the counts of otolith daily increments.

From the eastern North Pacific, archived otoliths of CA sardine captured in cruise surveys and in the pelagic fishery of the Southern California Bight during 1987, 1991–1998, and 2005–2007 were collected. Fish in the size range of 10–16 cm SL were regarded as age-1 individuals born in the previous year, following Takahashi and Checkley[56]. The number of individuals varied between year classes in the range of 4–20 (Supplementary Table 2).

### Otolith processing, microstructure and somatic growth analysis
Sagittal otoliths were cleaned to remove the attached tissue in freshwater and then air-dried. Otoliths of JP sardine were embedded in epoxy resin (Petropoxy 154, Burnham Petrographics LLC) on slide-glass, while those of CA were glued to slide-glass using enamel resin and then ground and polished with sandpaper to expose the core. For some otoliths of CA sardine, the polished surface was coated with additional resin to facilitate identification of the daily increment width. Using an otolith measurement system (RATOC System Engineering Co. Ltd.), the number and location of daily increments were examined

along the axis in the postrostrum from the core. Although daily increments were clearly observed until the otolith edge for JP sardine, it was difficult to do this for CA sardine probably because they had experienced winter when otolith growth slowed down. Therefore, the rings were counted as far as possible for CA sardine, which typically resulted in more than 150 counts. The first daily increment was assumed to form after 3 days post hatch (dph) for JP and 8 dph for CA sardine following Takahashi et al.[26] and Takahashi and Checkley[56]. The otolith radius at each age was calculated by adding all the increment widths up to that age. Standard lengths at each age were back-calculated assuming a linear relationship between otolith radius and standard length using the biological intercept method[34] as follows:

$$SL_n = \left( SL_{catch} - SL_{first} \right) \times \left( OR_n - OR_{first} \right) / \left( OR_{catch} - OR_{first} \right) + SL_{first}$$

(1)

where $SL_n$ is the standard length at age n, $SL_{catch}$ is the standard length at catch, $SL_{first}$ is the standard length at the age of first daily increment deposition fixed at 5.9 mm for JP sardine and 5.5 mm for CA sardine following the previous studies[26,56], $OR_n$ is the otolith radius at age n, $OR_{first}$ is the otolith radius at the age of first daily increment deposition, and $OR_{catch}$ is the otolith radius at catch. Based on rearing experiments of field collected eggs, Lasker[57] showed the SL of CA sardine at 6–8 dph ranged between 3.8 to 6.5 mm, and Matsuoka and Mitani[58] showed the total length at 2–4 dph ranged between 4.8 to 6.2 mm, corresponding to 4.7 to 6.1 mm in SL. To deal with these uncertainties regarding the size at the age of first daily increment deposition, we conducted Monte Carlo simulations (10,000 times) to estimate the uncertainties of back-calculated SL, assuming that the initial SLs fall between 3.8 to 6.5 mm for both sardines. Standard deviations of the temporal back-calculated SL at each age were presented as the uncertainty of each $SL_n$ estimation, which varied between 0.51 and 0.73 at the end of larval stage (JP: 45 dph, CA: 60 dph), between 0.34 and 0.64 at the end of early juvenile stage (JP: 75 dph, CA: 90 dph) and between 0.20 and 0.53 at the end of late juvenile stage (JP: 105 dph, CA: 120 dph). These values were significantly smaller than the variability of estimated SL among individuals assuming initial sizes of 5.9 and 5.5 mm for JP and CA sardine, respectively (standard deviations: 4.2, 8.1 and 8.3 in JP sardine and 5.5, 9.1 and 10.3 in CA sardine for the end of larval, early juvenile and late juvenile stages, respectively), suggesting that the back-calculated SL is robust to variations of initial size. Nevertheless, the biological intercept method assumes a constant linear relationship between fish and otolith size within individual[59], which can vary depending on physiological or environmental conditions[60,61]. Therefore, to examine the relationships between temperature and growth, we used both otolith growth, which contains fewer assumptions, and back-calculated somatic growth as growth proxies. Since the use of the two proxies did not show remarkable differences in the relationships between temperature and growth (Supplementary Figs. 11, 12), we mainly used the back-calculated SL in the discussion, which has a more direct ecological implication.

To more generally test whether growth trajectories are different between the western and eastern boundary current systems, otolith growth data of JP and CA sardines were compared with those of sardines in the east to south and west coasts of South Africa. The biological intercept method to back-calculate standard length could not be used in sardine from South Africa because the size at catch was large, some over 20 cm, and otolith radius and standard length were not linearly correlated for fish of this size. Therefore, the otolith radius and increment width were directly used as proxy for size and growth in this comparison, respectively. For visualisation (Fig. 2a), the means of year class mean otolith radii were estimated for JP and

CA sardines. For CA sardine, otolith radii at ages were simply averaged within each year class. For JP sardine, to account for the variation in the number of individuals captured at the same station, otolith radii were first averaged within each station, and the station means were averaged within each year, weighted by catch-per-unit-effort. For South African sardine, data of otolith daily increment widths from hatch to 100 dph of 67 adults captured at six stations on the east to south coast (>22ºE), and 51 individuals captured at six stations on the west coast (<20ºE) from 2015 to 2017, published in Sakamoto et al.[30], were used.

## Micro-milling, powder collection and isotope analysis

As the contamination of enamel resin used for CA sardine otoliths may potentially affect the isotope analyses, we removed the resin by dissolving it in acetone for the otoliths coated with the resin, and the otoliths were air-dried. They were then embedded in Petropoxy 154, which does not create any gas in the reaction with phosphoric acid, and ground again to expose the otolith surface. Test analyses using otoliths of horse mackerel demonstrated that this procedure (i.e., embedding with the resin and removing it with acetone) does not affect the subsequent isotope analyses. Before micromilling, the surface of the polished otolith was cleaned using an ultrasonic cleaner with Milli-Q water and air-dried for a few hours. The otolith portions deposited during hatch–30, 31–45, 46–60, 61–75, 76–90, 91–105, and 106–120 dph for the JP sardine and hatch–30, 31–60, 61–90, 91–120, and 121–150 dph for CA sardine, were milled out sequentially using a high-precision micromilling system Geomill 326. The difference in the temporal resolution was due to the slower growth rates of CA sardine. The milling depth was set to 50 μm for the area near the core and 100 μm for the rest. After each milling, the otolith was observed under a microscope to check for otolith fractions that had cracked outward from other milling area, and these were removed using a needle if present. The milled powders were then collected using a needle and a stainless-steel cup and poured into response vials. After each collection of powder, the otolith was cleaned with an air duster to avoid cross-contamination between the milling paths.

The $\delta^{18}O$ and $\delta^{13}C$ values of collected otolith powder were determined using the micro-volume analysis system MICAL 3c[62,63] at the National Institute of Technology, Ibaraki College, for the area nearest to the core, and the automatic system DELTA V + GAS Bench for the rest at the Atmosphere and Ocean Research Institute, the University of Tokyo. The otolith powders were reacted with phosphoric acid at 25 °C, and the released $CO_2$ was purified before being introduced into the mass spectrometer for the MICAL3c system. The response with phosphoric acid was performed at 72 °C for DELTA V. The $\delta^{18}O$ and $\delta^{13}C$ values were reported in δ-notation relative to the Vienna Pee Dee Belemnite (VPDB) scale and are given as a‰ value. Analytical precisions were better than ±0.10‰ and ±0.17‰ for $\delta^{18}O$, and better than ±0.10‰ and ±0.15‰ for $\delta^{13}C$, respectively. The acid fractionation factor of calcite was used to facilitate comparisons with isotopic values reported in previous studies[64]. Because the difference between the acid fractionation factor of calcite and aragonite is temperature dependent[65], 0.09‰ was subtracted from the $\delta^{18}O$ value determined by DELTA V to adjust for the different response temperatures.

## Estimation of $M_{oto}$

Dissolved carbon in fish blood is derived from two sources: dissolved inorganic carbon (DIC) from the ambient water and metabolic carbon released from the respiration of food. Hence, the isotopic composition of dissolved carbon in fish blood, and consequently in the otolith carbonate, is a weighted mean of the $\delta^{13}C$ values of DIC and metabolic carbon[66]. As the $\delta^{13}C$ values of DIC are generally higher than those of metabolic carbon, blood $\delta^{13}C$ values decrease when respiration rates increase and the proportion of metabolic carbon in blood increases[38]. The proportion of metabolically derived carbon in otolith carbonate

($M_{oto}$) from otolith $\delta^{13}C$ can therefore be used as a proxy for the field metabolic rate of fish. Following Chung et al.[38], $M_{oto}$ was estimated as:

$$M_{oto} = \left(\delta^{13}C_{oto} - \delta^{13}C_{DIC}\right) / \left(\delta^{13}C_{diet} - \delta^{13}C_{DIC}\right) + \varepsilon \quad (2)$$

where $\delta^{13}C_{diet}$ and $\delta^{13}C_{DIC}$ are the $\delta^{13}C$ values of the diet and DIC in seawater, respectively. The $\varepsilon$ term is the total net isotopic fractionation during carbon exchange between DIC and blood and between blood and endolymph in which the otolith is formed, which was set to 0 based on Solomon et al.[66] The $\delta^{13}C$ values of the diet of JP and CA sardines were estimated from the $\delta^{13}C$ of sardine muscle or zooplankton. The $\delta^{13}C$ values of JP sardine muscle reported in previous studies were in the range of −17.5 to −20.0‰ (−17.5 to −20.0‰ (Yasue et al.[67]: larvae and juveniles in the south coast of Japan); −18.4 ± 0.8‰ (Ohshimo et al.[68]: larvae to adults around Japan)), and $\delta^{13}C$ values of copepods in the Oyashio and Kuroshio-Oyashio regions were reported as −19.8 to −21.6‰[69]. Considering that the diet-tissue enrichment of $\delta^{13}C$ is approximately 1.5‰ in marine fish[70] we assumed that the $\delta^{13}C_{diet}$ for JP sardine would fall in the range of −19.0 to −22.0‰. Because the reported $\delta^{13}C$ values of the muscle of CA sardine in the California Current region were −17.0 to −20.0‰ (−19.8 ± 0.2‰[71]; −17.0 ± 0.8‰[72]; −17.5 to −18.0‰[73]), the $\delta^{13}C_{diet}$ for CA sardine was assumed to be in the range of −18.5 to −21.5‰. The $\delta^{13}C$ values of DIC in seawater were extracted from the World Ocean Database[74]. As the $\delta^{13}C_{DIC}$ is known to show temporal shifts due to the emission of anthropogenic $CO_2$ that results in a reduction of the $\delta^{13}C$ of atmospheric $CO_2$ known as the $^{13}C$ Suess effect[75], we extracted the $\delta^{13}C_{DIC}$ data observed in the Kuroshio-Oyashio system (130-180 °E, 30-45 °N) during 2006–2015 and the California Current system (110-130°W, 30-40°N) during 1986–2006. As the value varied between +0.53 and +1.05‰ in the former system and −0.31 and +2.20‰ in the latter system (Supplementary Fig. 2), we assumed that the $\delta^{13}C_{DIC}$ in each region fell within those ranges. To deal with these uncertainties regarding $\delta^{13}C_{diet}$ and $\delta^{13}C_{DIC}$, we conducted Monte Carlo simulations 10,000 times to estimate the mean and standard deviation of individual $M_{oto}$ values at the given life stages. The means of the $M_{oto}$ values were used in subsequent statistical analyses. Standard deviations of the temporal values were presented as the uncertainty of each $M_{oto}$ estimation, which varied between 0.01 and 0.03. The relationship between $M_{oto}$ and oxygen consumption rate has been determined only for juvenile Atlantic cod[38], although the relationship can be different between species and life stages especially for the larval stage when seawater DIC may be incorporated through cutaneous exchange rather than through the gills. Therefore, we did not convert $M_{oto}$ values to the oxygen consumption rate and analysed them directly as a metabolic proxy, and we compared them only within each life stage to investigate relationships between temperature and metabolic rate.

### Measurements of seawater $\delta^{18}O$ in the California Current region

As otolith $\delta^{18}O$ in fish is affected by temperature and $\delta^{18}O$ of ambient water, the seawater $\delta^{18}O$ distribution is essential for estimating the temperature from otolith $\delta^{18}O$. Although the distribution of seawater $\delta^{18}O$ in the western North Pacific has been relatively well studied (Supplementary Fig. 3a), data are limited in the eastern North Pacific especially off southern California where CA sardine grows. We therefore collected surface and sub-surface seawater samples for the isotope analysis that were collected during the 1708SR CalCOFI cruise survey conducted by the California Cooperative Oceanic Fisheries Investigations in August 2017. At every two stations on three line transects extending offshore from the Southern California Bight, seawater samples for $\delta^{18}O$ analysis were taken from 10 m and 50 m depths using CTD-attached Niskin bottles and preserved in sealed glass vials to prevent evaporation. The sampling range covered from the inshore area of the Southern California Bight to the

offshore California Current region which was assumed to represent larval and juvenile habitats off Southern California. After membrane-filtration (pore size: 0.45 μm, Toyo Roshi Kaisha, Ltd.), $\delta^{18}O$ values were measured at the National Institute of Technology, Ibaraki College using the Picarro L2130-i system. Data were reported in $\delta$-notation against the VSMOW (Vienna Standard Mean Ocean Water) with a precision better than ±0.05‰.

Seawater $\delta^{18}O$ of the Southern California Bight did not show large variation either horizontally or vertically (Supplementary Fig. 3b, c), ranging between −0.42 and −0.20‰, with mean value of −0.32‰ and a standard deviation of 0.05‰. The salinity of these samples measured by CalCOFI ranged between 33.08 and 33.56, and a significant correlation was detected between seawater $\delta^{18}O$ and salinity as follows:

$$\delta^{18}O = 0.279 \times Salinity - 9.63 \, (linear \ regression \ analysis,$$
$$n = 35, r^2 = 0.47, p = 4.7^* 10^{-6}) \quad (3)$$

The extent of potential inter-annual and seasonal variations in seawater $\delta^{18}O$ was analysed based on variations in salinity, as seawater $\delta^{18}O$ is generally correlated with salinity[76]. A total of 14732 salinity measurements of bottle samples in the upper 50 m of CalCOFI cruises in spring and summer during 1986, 1990–1997 and 2004–2006, which horizontally encompassed 117.2–125.7 °W and 29.8–35.1 °N, were extracted from the CalCOFI Hydrographic database (https://www.calcofi.org/ccdata/database.html, accessed on 17th April, 2020, currently changed to CalCOFI Data Portal (https://calcofi.org/data/)). Within the measurements, 69% (10194 measurements) were in the salinity range that occurred in our $\delta^{18}O$ measured samples, and 95% was in the range between 32.93 and 33.69, corresponding to −0.44 and −0.23‰ in seawater $\delta^{18}O$ based on the regression above (Eq. 3). This suggests that spatial and temporal variations of seawater $\delta^{18}O$ in the main habitat area of CA sardine are limited and likely to fall in the range of −0.32 ± 0.12‰.

### Conversion of otolith $\delta^{18}O$ to temperature

Otolith $\delta^{18}O$ is affected by both temperature and $\delta^{18}O$ of the ambient water. In the habitat area of CA sardine, the seawater $\delta^{18}O$ in the habitat area did not show large horizontal or vertical variation (Supplementary Fig. 3b, c), while seawater $\delta^{18}O$ in the habitat area of the JP sardine is known to show considerable variation, ranging from −1‰ to +0.5‰ (Supplementary Fig. 3a). Therefore, the temperatures were calculated using different methods.

As seawater $\delta^{18}O$ is generally correlated with salinity[76], otolith $\delta^{18}O$ can be regarded as a 2-variable function of temperature and salinity when seawater $\delta^{18}O$ variation cannot be ignored. If seawater temperature and salinity are completely independent of each other, it would be impossible to estimate both from just one otolith $\delta^{18}O$ value, although they are often closely related. Therefore, using the relationship between salinity and temperature, which varies annually and seasonally, estimating both parameters from otolith $\delta^{18}O$ would become possible with a certain range of error. To build formulas to calculate the temperature from otolith $\delta^{18}O$ for each month of each year, the surface layer (< 30 dbar in pressure) temperature and salinity observed by Argo floats in the range of 130–180 °E, 30–45 °N but excluding nearshore areas and the Sea of Japan, from February to October 2006–2010 and 2014–2015, were extracted from the Argo float dataset Advanced automatic QC(AQC) Argo Data ver.1.2a distributed by JAMSTEC[77] (Supplementary Fig. 4a). The number of observations in each month was approximately 2200 on average, varying from 319 to 5427. For each temperature and salinity pair, the corresponding otolith $\delta^{18}O$ was calculated using the seawater $\delta^{18}O$-salinity relationship in the Kuroshio-Oyashio system[39]

$$\delta^{18}O_{seawater} = 0.56 \times Salinity - 19.06 \quad (4)$$

and the otolith $\delta^{18}$O-temperature and seawater $\delta^{18}$O relationship for the JP sardine[37].

$$\delta^{18}O_{otolith} = \delta^{18}O_{seawater} - 0.18 \times Temperature + 2.69 \qquad (5)$$

The temperature was plotted against the otolith $\delta^{18}$O for each month, which generally showed a curve-shaped relationship (Supplementary Fig. 4b). A quadratic function was fitted to the plots using the least squares method and was used as the formula to estimate the temperature from otolith $\delta^{18}$O. The root-mean-square-errors of the formulas can be regarded as proxies for the accuracies of temperature estimation. These were 1.0 °C on average, varying from 0.3 °C to 2.0 °C with a tendency to increase in summer months (Supplementary Fig. 4b, c; Supplementary Table 3). All otolith $\delta^{18}$O values of the JP sardine were converted to temperature using the formula made for the month in which the median date of each milled area belongs. These analyses were performed using MATLAB R2017a (The MathWorks, Inc., Natick, Massachusetts, United States).

The ambient temperature of CA sardine was calculated using otolith $\delta^{18}$O-temperature and seawater $\delta^{18}$O relationship for the JP sardine[37] (Eq. 5), with the fixed seawater $\delta^{18}$O value of −0.32‰. As the seasonal and inter-annual variations of seawater $\delta^{18}$O were mostly limited to the range of ±0.12‰, the error due to these variations is smaller than 0.7 °C.

### Relationship between inter-annual variation of experienced water temperature and environmental indices

To understand how variations in habitat temperature are controlled, inter-annual variations of experienced water temperature were compared to environmental indices. As the distribution during juvenile stages can be significantly different between years due to the changes in dispersal and migration patterns[39], detecting the correlation between experienced water temperature in the stage and indices would be difficult. Thus, we focused on the temperature during 1 or 2 months from hatch. Year-class mean hatch dates of JP sardine varied between March and April. As there are no data for hatch date distribution for CA sardine, we assumed that CA sardine hatched in April, which is known as the peak spawning month of CA sardine (e.g., Lo et al.[78]). Based on these assumptions, the mean experienced temperature of JP sardine from hatch to 60 dph was compared to mean PDO[41] values from March to May, while the experienced temperature of CA sardine during 0-30 dph was compared to the PDO during April. The PDO index were downloaded from the webpage of National Centers for Environmental Information (https://www.ncei.noaa.gov/access/monitoring/pdo/, accessed on 10th August 2022). As we did not find significant correlations for CA sardine, the temperature for CA sardine was compared to the Coastal Upwelling Transport Index[42] (CUTI) off Southern California (mean of the indices at 33–36 °N). The CUTI index was downloaded from the MIKE JACOX webpage (https://mjacox.com/upwelling-indices/, accessed on 27th October 2020). It should be noted that because the number of data points (7 and 11 for JP and CA sardine respectively) are not large, the relationships observed here may not represent the true relationship between sardine nursery temperature and environmental indices, even if correlation coefficients may be high.

### Statistical analyses for the differences of otolith increment widths among regions

To understand the mechanism creating the differences and similarities in otolith growths trajectories among JP, CA, and west coast, and south-east coast SA sardines (Fig. 2a), otolith increment widths during every 10 days between hatch and 100 dph were analysed (Supplementary Fig. 5). A linear mixed-effects model based on the R 4.1.3 and the libraries lmerTest, MuMIn and emmeans was used. Each 10-day increment width (IW) was modelled by two fixed factors (Region and

Age) and a random effect (individual fish, Fish.ID) as lmer($IW \sim Age*Region + (1 \mid Fish.ID)$). Here, age was used as a factor. The diagnostic for the model showed a straight Q-Q plot and the normality of residuals (Supplementary Fig. 6). The pairwise comparison between regions at each age using *emmeans* showed that there were significant differences in some pairs in most of the age range, although increment widths of JP and the south-east coast SA sardines were not significantly different during 21–50 dph and those of CA and the west coast SA sardines were not significantly different during 31–60 dph (Supplementary Table 4).

### Statistical analyses for the differences of $M_{oto}$ and experienced seawater temperature between JP and CA sardines

To test for the differences in $M_{oto}$ and experienced seawater temperature between JP and CA sardines, we used a linear mixed-effects model based on the R 4.1.3 and libraries lmerTest, MuMIn and emmeans. The $M_{oto}$ and temperature were modelled by two fixed factors (Region and Age) and a random effect (individual fish, Fish.ID) as lmer($M_{oto} \sim Age*Region + (1 \mid Fish.ID)$) and lmer($Temperature \sim Age*Region + (1 \mid Fish.ID)$), respectively. Outliers of $M_{oto}$ or the temperature detected using the *boxplot* function were excluded in each analysis. Here, age was used as a factor and values for every 15-day interval for JP sardine were grouped into the age groups for every 30-day interval for the test (0–30, 31–60, 61–90, 91–120 dph). The diagnostics for the models showed mostly straight Q-Q plots and normalities and homogeneities of the residuals (Supplementary Figs. 7, 8). The pairwise comparisons showed that $M_{oto}$ was significantly higher in JP sardine in all age groups and the experienced temperature was significantly higher in age groups except for 91–120 dph than in CA sardine (Supplementary Tables 5, 6).

### Statistical analyses for the effect of temperature on $M_{oto}$ and $M_{oto}$ variations

The relationship between the experienced water temperature and $M_{oto}$ was investigated to test the effect of temperature on metabolic performance. As $M_{oto}$ was strongly dependent on life stage (Fig. 2b), we used life-stage-averaged data for each individual (JP larva: 0–45 dph, early juvenile: 46–75 dph, late juvenile: 76–105 dph, CA larva: 0–60 dph, early juvenile: 61–90 dph, late juvenile: 91–120 dph). Outliers of $M_{oto}$ detected using the *boxplot* function in R 4.1.3 for each stage and region were excluded.

First, to test the difference of $M_{oto}$ between JP and CA sardines taking the effect of temperature into account, we applied a generalised linear model. Gaussian function family function was used as overall $M_{oto}$ distribution showed no significant discrepancy from Gaussian distribution (Shapiro-Wilk test, w = 1.00, $p > 0.05$). Theoretically, the relationship between metabolism and temperature tends to show a linear trend after the metabolic rate is log-transformed[79]. Thus, we applied "identity (data without transformed)" and "log (data transformed)" links to evaluate if model shows a better linearity with data transformation. Based on AIC, however, the result showed $M_{oto}$ have a better linearity without data transformation (Supplementary Table 7). We, therefore, used "identity" links for the further model selection. Model selection base on AIC was performed for models including temperature, region (JP and CA sardines), life history stages (larvae, early juvenile and late juvenile) and interactions of these factors. The full model including all the interactions had the lowest AIC (Supplementary Table 7). As the diagnostic for the full model showed normality and homogeneity of residuals (Supplementary Fig. 9), we selected this model for interpretation. The CA sardine at the larval stage as the baseline, we found only JP sardine at early and late juvenile stages has relatively higher $M_{oto}$ values, and the temperature-dependent slope is significantly gentler in JP sardine at early and late juvenile stages (Supplementary Table 8).

Next, the diversity of $M_{oto}$ across temperature range was assessed to estimate the optimal temperature in each stage. The relationship between the maximum metabolic rate and temperature is known to be parabolic, while that between the standard metabolic rate and temperature is logarithmic[28,79]. As the highest field metabolic rate would be constrained by maximum metabolic rate and the lowest field metabolic rate would be close to resting metabolic rate[43], fish would have the most diverse metabolic performance at the optimal temperature with the widest aerobic scope. Thus, we modelled the highest and lowest $M_{oto}$ values in each 1 °C bin using a polynomial regression and a generalised linear model with Gaussian distribution and a log link for the 95th and 5th percentile values of each bin, respectively (Supplementary Fig. 10). The values of the bin that included less than four values were excluded from the regression analyses to reduce the uncertainty caused by under-sampled temperature bins. The gap between the two regression lines was considered as a proxy for the aerobic scope, and the temperature at which the gap reached the maximum was regarded as the optimal temperature.

### Statistical analyses for the relationships between temperature and growth

To understand how variation in ambient water temperature affects early life growth of sardines, we compared back-calculated standard length at around the end of the larval stage (hatch–35 mm; JP: 45 dph, CA: 60 dph), the end of the early juvenile stage (35–60 mm; JP: 75 dph, CA: 90 dph), and the end of the late juvenile stage (60–85 mm; JP: 105 dph, CA: 120 dph) and the mean seawater temperature from hatch to the ages. Median of each sampling batch were used as minimal data unit. Pearson's r and p-values were first calculated for each comparison (Supplementary Table 9). As the relationship between mean temperature and standard length of JP at 75 dph seemed to be dome-shaped rather than linear, we introduced quadratic term of temperature and tested whether the term increased explanatory power using a linear model and stepwise model selection based on AIC. The model selection showed that the full model (*Standard length ~ Temperature$^2$ + Temperature*) was the best model, and the coefficients of the quadratic and linear terms were both significant (Supplementary Table 10). To account for these multiple tests, we corrected the p-values of the coefficients of the quadratic term in the linear model for JP sardine at 75 dph and of the Pearson's r for the rest using the Benjamini-Hochberg procedure with $\alpha = 0.05$, and selected the null hypotheses that could be rejected (Supplementary Table 9). To compare the temperature that allow maximisation of growth rate and optimal temperature derived from the analysis of $M_{oto}$ for each stage, median somatic growth rate and otolith increment width in each 1 °C bin was calculated together with its 3-window running mean (Supplementary Figs. 11, 12).

### Statistical analyses for the relationships between sea surface temperature and survival index

To test whether habitat temperatures during the first 4 months after hatch affect the survival of sardines in the first year of life on a multi-decadal scale, satellite-derived sea surface temperature (SST) since 1982 and survival of JP and CA sardines were compared. The log recruitment residuals from Ricker recruitment models (LNRR)[13], representing early life survivals taking into account the effect of population density, were calculated based on the stock assessment data for JP and CA sardines as follows:

$$LNRR_t = ln(R_t/S_t) - (a + b \times S_t) \qquad (6)$$

where $LNRR_t$ is the LNRR at year $t$, $R_t$ is the recruitment of year-class $t$, $S_t$ is the spawning stock biomass in year $t$, and a and b are the coefficients of linear regression of $ln(R_t/S_t)$ on $S_t$. Pearson's r between the LNRR and the mean SST values from March to June

for JP and from April to July for CA sardine was calculated for each grid points in the western and eastern boundaries of the North Pacific basin, derived from a SST product based on satellite and in situ observations[80] (Global Ocean OSTIA Sea Surface Temperature and Sea Ice Reprocessed (https://resources.marine.copernicus.eu/product-detail/SST_GLO_SST_L4_REP_OBSERVATIONS_010_011/INFORMATION), accessed on 11th August and 28th October 2021). The correlations were generally negative and positive in the western and eastern regions, respectively (Supplementary Fig 13a, b). In particular, mean SST values in the area where eggs, larvae and juveniles of JP or CA sardines are mainly found in the months[26,39,49,56,78,81,82] (dotted areas in Supplementary Fig 13a, b) were compared with LNRR values to test the relationship between habitat temperature and survival in the early life stages (Supplementary Fig 13c). It should be noted that the mean SST values were not significantly correlated with otolith-derived year-class mean temperatures of JP and CA sardines during the larval to late juvenile stages (JP: r = 0.01, p = 0.98, n = 7, CA: r = 0.29, p = 0.38, n = 11), likely due to the short periods analysed, patchy distribution and inter annual variation in larval and juvenile dispersal and migration patterns. Nevertheless, the regions included areas where SST showed weak to significant ($p < 0.05$) positive correlations with otolith-derived temperatures (Supplementary Fig 13a, b). In addition, mean SST values for the western and eastern regions were significantly correlated with PDO (1982–2016, March to June, r = −0.56, $p < 0.001$, n = 35) and Coastal Upwelling Transport Index (31–40 °C, 1988–2016, April to July, r = −0.50, $p < 0.01$, n = 29) in longer time scales, respectively. These are the indices that were able to reasonably explain otolith-derived temperatures during hatch to 30–60 dph (Fig. 2d, e). These suggest that the mean SST values in the regions at least partly reflect the habitat temperature of the sardines and can be used to test the relationship between habitat temperature and survival.

### Reporting summary
Further information on research design is available in the Nature Research Reporting Summary linked to this article.

## Data availability
The otolith and seawater isotope ratio and otolith microstructure data generated in this study have been deposited in the Dryad under accession code https://doi.org/10.5061/dryad.5mkkwh78j. PDO index data can be downloaded from https://www.ncei.noaa.gov/access/monitoring/pdo/. CUTI data can be downloaded from https://mjacox.com/upwelling-indices/. The data of stable carbon isotope ratio of DIC can be downloaded from https://www.ncei.noaa.gov/products/world-ocean-database. CalCOFI hydrographic data can be downloaded from https://calcofi.org/data/.The Argo float data can be downloaded from http://www.jamstec.go.jp/e/database/. The sea surface temperature products can be downloaded from https://resources.marine.copernicus.eu/product-detail/SST_GLO_SST_L4_REP_OBSERVATIONS_010_011/INFORMATION. Source data are provided with this paper.

## Code availability
The scripts to analyse data and generate the graphs are publicly available on Zenodo repository https://doi.org/10.5281/zenodo.6983520.

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

## Acknowledgements

We thank Noriko Izumoto (Atmosphere and Ocean Research Institute, The University of Tokyo) for contributions to the $\delta^{18}O$ analysis of otolith powders. We also appreciate Yasuhiro Kamimura, Chikako Watanabe (Japan Fisheries Research and Education Agency), Barbara Javor (Southwest Fisheries Science Center, NOAA), and Dianna Porzio (California Department of Fish and Game) for providing archived otolith samples, and Jennifer Rodgers-Wolgast, Ralf Goericke, and David Wolgast (Scripps Institution of Oceanography, University of California San Diego) for providing seawater samples in the California Current system. We are also grateful to David Checkley (Scripps Institution of Oceanography, University of California San Diego) for providing comments to improve our manuscript. This work has been conducted as a part of the Ph. D. thesis of T.S., although the main text and analyses of the data are thoroughly modified. This study was funded by the Research Fund KAKENHI Grants from the Japan Society for the Promotion of Science (JSPS) to T.S. (17J00556), M.T. (22780185), K.S. (22H05026, 22H05028) and T.I. (16H02944, 18H04921).

## Author contributions

Conceptualization: T.S., M.T., Methodology: T.S., M.C., K.S., T.I., T.H., Investigation: T.S., K.S., T.I., T.H., Visualization: T.S., M.C., Funding acquisition: T.S., M.T., T.I., Project administration: K.K., M.T., Supervision: M.T., K.K., Writing—original draft: T.S., Writing—review & editing: R.R., M.T., M.C., K.S., T.I., T.H.

## Competing interests

The authors declare no competing interests.
