## [Peer Review File · Nature Communications]

Contrasting life-history responses to climate variability in eastern and western North Pacific sardine populationsREVIEWER COMMENTS

Reviewer #1 (Remarks to the Author):

This manuscript takes a sophisticated approach to investigating the growth dynamics of sardines, seeking at the most fundamental level why subpopulations with similar life histories respond differently to large-scale environmental conditions. The manuscript considers an important ecological and economic group (sardines) and one that is often used to represent the dynamics of important marine systems (eastern and western boundary currents). Otolith proxies of individual growth, temperature experienced, and field metabolic rate are used to explain the puzzle of opposite responses of sardines in the eastern and western Pacific. The work is convincing and important, and relevant in the fields of fisheries oceanography, climate effects, ocean productivity, and trophic modelling. The link to optimal temperatures and field metabolic rates are relevant to physiological ecology.

There are also several weaknesses in the manuscript in the current form:

- 1) While it is logical to try to extend the E-W boundary current/sardine comparisons from the Pacific to South Africa, this isn't done consistently enough in the manuscript to convince the reader. The focus is almost entirely on the Pacific populations, and only once is there any reference to the SA sardines. It is also confusing to see these as part of Figure 1, before anything is even mentioned about looking wider to see how general the E-W comparison would be. There is a lot of merit in adding more E/W pairs, but here it just seems to be tacked on and not integrated into the main message. This can be solved by either expanding to include the SA (and any other pairs) results better into the text, or by eliminating them entirely - except perhaps by suggesting that these patterns may hold more universally.**
- 2) Why are the growth comparisons done with a complex grid of t-tests rather than a more usual mixed modelling approach?**
- 3) It would be very useful for the species names to be given at some point in the text**
- 4) what portion of the life-span is represented by the life history stages that are analysed here? And how much is overall growth and population dynamics reliant on growth during these early stages**
- 5) It isn't exactly clear why otolith growth (or increment widths) isn't used directly as a growth proxy, rather than back-calculating. This would avoid any errors in assumptions of the backcalculation model. If that is used for the SA sardines, then why not for the Pacific sardines?**
- 6) A short explanation for the different age bands used for JP and CA would help general readers (assuming this is because of slower growth and lower temperatures in CA) - and here, does the different amount of otolith material in JP vs CA have the potential to affect the results in any way?**

Reviewer #2 (Remarks to the Author):

**Otolith disentangles East-West puzzle in population dynamics of sardines
Sakamoto et al.
Nature Communications**

Summary

This manuscript uses otolith-derived estimates of growth and metabolic proxies to distinguish unique responses of sardine populations of the western and eastern North Pacific to temperature fluctuations and basin-scale climate indices. There is a considerable amount of work included in this manuscript, and the authors should be proud of how clearly they were able to convey their message given the scope of work. However, there are still a few ways this manuscript can be improved. I have included specific comments below (e.g., on methods, analyses, etc.), but in general I think this manuscript will be of interest to the fields of ecology, climate science, and marine resource management, especially given the potential for the findings to be used in more

directed studies which employ population models to elucidate the population-level effects of growth response to environmental change (e.g., temperature, climate). Having said that, the conclusion and discussion section generally lacks a paragraph on how this work matters beyond ecology. How can the findings be used in subsequent work to inform resource management of sardine populations and fisheries which harvest them? What does the difference in response between basin-specific populations potentially mean for the fate of those populations, given how temperatures will continue to rise and climate regimes may continue to change in the future? These are the broad-scale discussion points I think this work leads into, and a paragraph discussing this in the context of current literature may provide a more comprehensive ending to the study and where it leads to next. There is extensive work on distributional shifts as a result of climate change, and the breadth of that work somewhat overshadows the effects we may see with respect to life history changes. However, I think this manuscript could be a catalyst to motivate further study on the potential effects on population-level life history changes moving forward if desired.

Specific Comments

- The title seems a little off and ambiguous – east-west of which ocean basin, a single otolith, etc.? Maybe “Otoliths distinguish Eastern vs. Western populations of North Pacific sardine using variable life history responses to climate change” instead? ‘Disentangles’ suggest these are mixing populations, but I don’t believe that is the case.
- After reviewing the paper, I think it might be a good idea to remove or reword the statement about prey availability in the abstract. This was not specifically studied in the submitted work. Stick to what was studied (metabolic response to environmental change, i.e., temperature and PDO/CUTI), and simply state that prey availability may have compounded the issue based on findings from other works across the field. I also think it may be a stretch to say this work has explained sardine population fluctuations. It certainly investigates how climate has influenced growth and metabolic proxies across sardine populations, but to investigate fluctuations in population abundance or biomass estimates a population model would be required. It may be better to say that the results could contribute to understanding population fluctuations, but more work is necessary to integrate the findings of the current study into a population modelling framework.
- [Lines 64-67] Can the authors more transparently distinguish western and eastern coasts vs. boundary currents? I understand what they mean, but a reader may associate the eastern boundary current with the south/east coast, whereas I believe it should correspond to the western coast. A simple parenthetical note relating coasts and associated currents should suffice.
- [Lines 75-76] Please specify inner ear or vestibular system.
- [Line 79] Citing the author’s name and including the superscript citation number seems redundant (“Hoie et al.¹⁶”, especially when next to a citation that does not include this (17)).
- [Line 114] Why use a parenthetical citation here (19) instead of superscript?
- [Lines 115-117] Can the authors report if the South African samples only consisted of individuals that lived to 100 dph, or if this is just the timespan that growth histories matched JP and CA samples? If the latter, it is probably a good idea to plot all of the data (>100 dph) so the reader can see how growth histories deviate from JP and CA samples after 100 dph.
- I think it needs to be stressed somewhere in the main text that Moto is a proxy, not a measurement, as it is being derived from previously published data on d13Cdiet and d13CDIC, not measured through samples collected by the authors in parallel to otolith sampling. Therefore, inferences made corresponding to Moto are based on assumptions and post-hoc treatment of previously published data, and may not exactly reflect reality.
- [Line 152] ANCOVA requires linearity of the regressions, and the polynomial regression relationships of 95th & 5th percentiles of Moto plotted in Figure 3a, 3d, and 3g do not all look linear. It’s also hard to decipher linearity vs. nonlinearity in the Extended Data Figure 5 of the Appendix. If the authors have violated ANCOVA’s assumption of linearity, they may need to find an alternative test or method to use (e.g., maybe AIC model selection exercise?). Or perhaps the regression of all data approaches

linearity. If so, please plot the linear regression for each area as a dotted line colored according to area.

- [Line 163, Figure 3] Moto requires a subscript on oto. Also, 'Juvenile I' is not a useful moniker and kind of clunky after "Early Juvenile" – perhaps "Late Juvenile" depending on what the authors are trying to convey? In (a), (d), and (g), please make the JP lines red instead of black to match the rest of the figure.
- [Line 184, Figure 4] The sentence starting "Significant regression lines.." in the caption is incomplete.
- [Lines 186-196] The authors discuss the importance of variability in survival and growth on recruitment success, and make a conclusion about population increases relying on cooler vs. warmer temperatures in western vs. eastern Pacific populations. However, I see no data integrated into the analysis on population abundance, population biomass, or year-class strength matriculating through these populations through time. Therefore, I am not sure how a manuscript analyzing growth, metabolism, and the influence of temperature on these processes, can make a definitive conclusion on population increase. I do see that temperature data is overlain on the plot of SSB in Figure 1, however, this is by no means sufficient to make generalizations on the influence of temperature on SSB. This could be more formally addressed using temperature or climate indices as environmental data within population models.
- The critique I made in the previous bullet point leads me to think about the title of this manuscript. Are the authors truly disentangling a puzzle in population dynamics? Growth models are certainly a central function in population dynamics models, but I think the use of this term in the title leads the reader to believe there will be more to the analysis than simply investigating how temperature influences larval and juvenile growth in different populations of sardine across the North Pacific Ocean. A more comprehensive integration of this new information on growth and metabolism into population models is suggested with the current title, and since that is lacking I suggest re-focusing the title on the content that is covered in the manuscript: temperature and climate effects on growth and metabolic proxies.
- [Lines 209-219] The authors discuss how the optimal temperatures (green bars) correspond to the individuals with the largest sizes in each JP case (Fig 4a,c,e), but certainly not in the CA case. This sends a mixed message that optimal temperature may not be a universally important determinant of optimal growth. Alternative to the authors' discussion on the additional importance of prey availability, this disconnection could be related to the assumptions made in generation of Moto, which would suggest that the estimates of Moto may not be the best for the CA region. This discussion on the discontinuities in main findings between study regions should be in the main text.
- [Line 210] Is Figure 4 actually showing slower or faster growth relative to optimal growing temperature, or smaller and larger sizes relative to this? I think this is an important distinction that needs to be made throughout the manuscript. Otolith increment width conveys growth rate, as does the slope (or growth coefficient parameter) of a size-at-age relationship. However, the relationship between standard length and temperature is plotted, and there is no indication of a change in slope in this relationship at temperature below or above the optimal (green box), I believe the sentence I reference here is talking about size, not growth (i.e., "largest sizes of CA sardine were consistently larger at temperatures above the optimal temperature"). The authors may want to adjust the conclusions on growth made in this paragraph to account for this distinction.
- [Line 227] 'change' should be 'changing' I believe.
- [Line 232] Again, size-at-age, growth, and population biomass increases are all very different, and I'm not sure how this manuscript can make definitive conclusions on population-level increases in biomass or abundance based on temperature effects on size-at-age, growth, and metabolism. Usually that requires a population model or stock assessment model.

Reviewer #3 (Remarks to the Author):

The authors produced quite an interesting study to compare two populations of sardines

based on growth increment width, and stable isotope analysis of otoliths that could reveal the level of metabolism © and ambient temperatures (O). However, the authors did not study the survival of larvae at different temperatures. Hence their conclusions about determinants of population growth of sardines (in terms of biomass) are not correct. Based on their data, the authors could definitely speak about differences in individual growth rates, but not in population growth. Then I wonder if they could achieve their goal to 'disentangle the puzzle' of biomass variability of sardines at both sides of the North Pacific.

I would suggest that the authors need to bring the evidence of survival variability, or downgrade their goal to study individual growth of both populations. In both cases the ms requires major revision.

Specific comments:

Lines 32-34. I don't think that you have studied survival of larvae and small juveniles here relative to temperatures? In this case how can you judge the fluctuations of biomass based just on individual growth rates?

Lines 36-95. The introductory part of the manuscript should be re-organised, as not all readers are familiar with sardines and their fluctuations in abundance. Start first with description of sardines and their life cycles (lines 91-93), and put much more detail into it – are Japanese and Californian sardines the same species? If yes, what are the connections between populations? What kind of abundance variability is characteristic to each population? And then, how you suggest to solve it – by looking at metabolism signatures in their otoliths.

Lines 97-100. Please describe the sampling scheme in a bit more detail. You collected sardines from JP and CA in different years – why? How can you compare them in you collected them in different years, especially those aged 0+ and 1? I would like to see justification for that. I guess if you the goal was to sample populations only during biomass increases – you need to have the data during biomass decrease to compare? You have to also specify how did you estimate the age of sardines – by counting daily growth increments? Are they validated?

Lines 101-102. You have to proof that the increment width in the otolith corresponds well with growth rates of the whole fish. Sometimes it is a good correlation, but sometimes it is not as the otolith growth is not directly correlated with amount of food consumed and transformed into body weight.

Line 107 – as above – you have to show that you have a good estimate of 'somatic growth' using otolith growth increment widths.

Line 113. What are 'otolith radii'?

Lines 115-117. What did you mean by 'analogous', and why it is a surprise to you?

Line 126. Even if JP sardines live in the warmer environment and hence have higher metabolic rates, it does not necessarily mean that they have higher growth rates. You need to prove it or to present the evidence here (like optimum temperatures for juvenile growth).

Lines 180-182. I did not understand correctly what did you mean by 'mean growth from hatch to the end of juvenile stage'. How did you calculate that? Dividing mean length at the end of the juvenile stage by the DPH?

Lines 191-196. How do you correlate the individual growth of fish (somatic growth) with the growth of population (which is somatic growth + survival rate)? I don't think that you presented here any data on increased survival of either population of sardines at different temperatures?

Lines 228-232. I don't think that you did a correct conclusion here. Even if your findings are correct, that means that sardines of both populations grow differently, but it does not explain their population growth! You need to discuss that.

Reply to the reviewer comments

We are truly grateful to the reviewers for their beneficial comments and suggestions on our manuscript. The comments have helped us to improve the paper considerably. Our replies to the comments are listed below in due order with red font. The concern most stressed by the reviewers was whether the response of early life growth to environmental variability can explain the responses of the populations. We agree that this is an important point. However, there is a rich literature showing strong associations between early life stage growth and survival or recruitment in various fish taxa, including sardines in the regions we studied. Furthermore, we provide additional data showing that sea surface temperatures around the main habitat of JP and CA sardine during spring to summer were negatively and positively correlated with their early life survival index, respectively. This is consistent with a number of studies linking habitat temperature to sardine production and suggests that the relationship between temperature and growth we observed is connected to population dynamics of the sardines. Therefore, our results support the hypothesis that faster individual growth promoted by climate change is of critical influence of population growth. We, therefore, believe that our analyses of the response of growth and metabolic proxy to climate change address the core of the puzzle of population dynamics of sardines. Although these relationships can address the mechanisms driving different population responses to changing environmental conditions, we also recognise that these are only components of the information necessary to assess the population dynamics. Hence, we have substantially revised the manuscript to emphasise these points, to relax the statements concerning population dynamics and to address all other concerns.

REVIEWER COMMENTS

Reviewer #1 (Remarks to the Author):

This manuscript takes a sophisticated approach to investigating the growth dynamics of sardines, seeking at the most fundamental level why subpopulations with similar life histories respond differently to large-scale environmental conditions. The manuscript considers an important ecological and economic group (sardines) and one that is often used to represent the dynamics of important marine systems (eastern and western boundary currents). Otolith proxies of individual growth, temperature experienced, and field metabolic rate are used to explain the puzzle of opposite responses of sardines in the eastern and western Pacific. The work is convincing and important, and relevant in the fields of fisheries oceanography, climate effects, ocean productivity, and trophic modelling. The link to optimal temperatures and field metabolic rates are relevant to physiological ecology.

-Response: We thank the reviewer for the positive and helpful comments.

There are also several weaknesses in the manuscript in the current form:

1) While it is logical to try to extend the E-W boundary current/sardine comparisons from the Pacific to South Africa, this isn't done consistently enough in the manuscript to convince the reader. The focus is almost entirely on the Pacific populations, and only once is there any reference to the SA sardines. It is also confusing to see these as part of Figure 1, before anything is even mentioned about looking wider to see how general the E-W comparison would be. There is a lot of merit in adding more E/W pairs, but here it just seems to be tacked on and not integrated into the main message. This can be solved by either expanding to include the SA (and any other pairs) results better into the text, or by eliminating them entirely – except perhaps by suggesting that these patterns may hold more universally.

-Response: We appreciate the reviewer for this point and agree that the comparison with SA was not sufficiently integrated to our conclusions. Based on the suggestion, we further integrated the comparison with SA in the discussion. First, we described that sardines are distributed in all temperate oceans except the western Atlantic in the first paragraph to show sardines generally inhabit the western and eastern boundaries of ocean basins (Line 41-45). Next, based on the similarity of differences in life history traits of JP-CA and south-west SA sardines, we suggested that the differences may reflect different environments of E-W systems (Line 165-169). Furthermore, we mentioned that the mechanism connecting temperature and population growth observed in JP and CA sardine may work more universally for other western and eastern boundary current systems (Line 252-255), and assumed as such in the future projection of sardine populations (Line 321).

2) Why are the growth comparisons done with a complex grid of t-tests rather than a more usual mixed modelling approach?

-Response: We agree to the reviewer that alternative statistical approaches could have been taken. However, otolith growth curves here follow Gompertz growth model. Because of its nonlinearity, model fitting including random effects is not necessarily straightforward, and interpretation of the result become complicated, if not impossible. As the aim here is to detect significant differences of size at ages between regions rather than to estimate parameters of growth curves, we feel that ANOVA and t-tests at every 10 days with correction for multiple tests are sufficient.

3) It would be very useful for the species names to be given at some point in the text

-Response: The (sub)species names were added when introducing sardine populations in each region (Line 57,58, 93-94). *Sardinops* has shallow genetic divergence and is considered to be comprised of several subspecies, rather than species (Grant and Bowen, 1998, Line 42).

4) what portion of the life-span is represented by the life history stages that are analysed here? And how much is overall growth and population dynamics reliant on growth during these early stages

-Response: The period we analysed (120-150 days post hatch) includes the entire larval and most of the juvenile stages that likely cover the period during which variations in mortality rates play a great role in population regulation and determination of year-class strength (Sogard, 1997, Line 130-133). Takahashi et al., (2008) found strong correlation ($R^2 > 0.8$) between recruitment and somatic growth around the early juvenile stage in JP sardine during 1996-2003. Although the strength of the correlation temporally varied when analysed for longer periods (Furuichi et al., 2020), the positive effect was consistently evident, suggesting that recruitment and population fluctuation are significantly related to growth rates during early life stages (Line 82-85).

5) It isn't exactly clear why otolith growth (or increment widths) isn't used directly as a growth proxy, rather than back-calculating. This would avoid any errors in assumptions of the backcalculation model. If that is used for the SA sardines, then why not for the Pacific sardines?

-Response: Although otolith and body size are closely correlated, studies have shown that otolith: size ratio varies systematically with somatic growth rates, resulting in relatively large otoliths in slow-growing fish (fish in general: Campana, 1990; for JP sardine: Takasuka et al., 2008). Therefore, using the otolith radius or increment width directly as a growth proxy would result in underestimation of the difference between faster and slower growing groups. As the back-calculation algorithm (the biological intercept method) that uses sizes of fish and otolith at catch and assume individual-specific linear relationships between otolith and fish size has been shown to be capable of eliminating this bias, it was ideal to use back-calculation also for the SA sardines to accurately compare somatic growth histories. However, as described in Methods (Line 412-415), sardines from South Africa were large, some over 20 cm, and otolith radii and standard lengths were not linearly correlated for fish of this size, which prevented us from the use of back-calculation. Because of this we used otolith radius for comparison among the Pacific and SA population with slight compromise, and backcalculated length for temperature-size comparisons of the Pacific populations.

6) A short explanation for the different age bands used for JP and CA would help general readers (assuming this is because of slower growth and lower temperatures in CA) - and here, does the different amount of otolith material in JP vs CA have the potential to affect the results in any way?

-Response: The difference in the temporal resolution was due to the slower growth rates of CA sardine as described in Methods (Line 438-439). As isotope values analysed by mass spectrometry are generally dependent on the mass of material, reference materials of multiple weight levels that cover the weight range of the samples are always analysed together with samples and used to adjust the raw values of samples. Therefore, the different amount of otolith material does not affect isotope values.

In the downstream analysis of the relationships among growth, metabolism and temperature based on the isotope values, we consistently used life-history stage mean values. This procedure allowed the analyses of ecologically consistent data with similar temporal resolution. Thus, we believe that our findings were not affected by the use of different age bands for JP and CA sardine.

Reviewer #2 (Remarks to the Author):

Otolith disentangles East-West puzzle in population dynamics of sardines

Sakamoto et al.

Summary

This manuscript uses otolith-derived estimates of growth and metabolic proxies to distinguish unique responses of sardine populations of the western and eastern North Pacific to temperature fluctuations and basin-scale climate indices. There is a considerable amount of work included in this manuscript, and the authors should be proud of how clearly they were able to convey their message given the scope of work. However, there are still a few ways this manuscript can be improved. I have included specific comments below (e.g., on methods, analyses, etc.), but in general I think this manuscript will be of interest to the fields of ecology, climate science, and marine resource management, especially given the potential for the findings to be used in more directed studies which employ population models to elucidate the population-level effects of growth response to environmental change (e.g., temperature, climate). Having said that, the conclusion and discussion section generally lacks a paragraph on how this work matters beyond ecology. How can the findings be used in subsequent work to inform resource management of sardine populations and fisheries which harvest them? What does the difference in response between basin-specific populations potentially mean for the fate of those populations, given how temperatures will continue to rise and climate regimes may continue to change in the future? These are the broad-scale discussion points I think this work leads into, and a paragraph discussing this in the context of current literature may provide a more comprehensive ending to the study and where it leads to next. There is extensive work on distributional shifts as a result of climate change, and the breadth of that work somewhat overshadows the effects we may see with respect to life history changes. However, I think this manuscript could be a catalyst to motivate further study on the potential effects on population-level life history changes moving forward if desired.

-Response: We thank the reviewer for a number of encouraging and insightful comments. We agree that the manuscript lacked discussion on how our results matters in broader context. A paragraph describing how sardine populations will be affected by global warming, together with implications for

evolutional strategies of sardines was added following the suggestion (Lines. 299-327).

Specific Comments

- The title seems a little off and ambiguous – east-west of which ocean basin, a single otolith, etc.? Maybe “Otoliths distinguish Eastern vs. Western populations of North Pacific sardine using variable life history responses to climate change” instead? ‘Disentangles’ suggest these are mixing populations, but I don’t believe that is the case.

-Response: We agree with the reviewer that mixing of populations is not the topic here. Although the proposed title seems nice, it should be 15 words and fewer according to the author guideline. To avoid questions posed by the reviewer, we modified the title to **“Contrasting life-history responses to climate variability in eastern and western North Pacific sardine populations”**, extending the reviewer’s idea.

- After reviewing the paper, I think it might be a good idea to remove or reword the statement about prey availability in the abstract. This was not specifically studied in the submitted work. Stick to what was studied (metabolic response to environmental change, i.e., temperature and PDO/CUTI), and simply state that prey availability may have compounded the issue based on findings from other works across the field. I also think it may be a stretch to say this work has explained sardine population fluctuations. It certainly investigates how climate has influenced growth and metabolic proxies across sardine populations, but to investigate fluctuations in population abundance or biomass estimates a population model would be required. It may be better to say that the results could contribute to understanding population fluctuations, but more work is necessary to integrate the findings of the current study into a population modelling framework.

-Response: We agree with the reviewer that prey availability was not specifically studied here. The statement regarding the effect of prey availability is, therefore, weakened to “Varying metabolic response to environmental changes, perhaps together with the dissimilarity in prey availability in different ecosystems described in previous studies, have led to contrasting growth responses.”. The answer to the latter concern is described in our next comment.

(Moved from later parts as these are well related to the comment above)

- [Lines 186-196] The authors discuss the importance of variability in survival and growth on recruitment success, and make a conclusion about population increases relying on cooler vs. warmer temperatures in western vs. eastern Pacific populations. However, I see no data integrated into the analysis on population abundance, population biomass, or year-class strength matriculating through these populations through time. Therefore, I am not sure how a manuscript analyzing growth, metabolism, and the influence of temperature on these processes, can make a definitive conclusion on population increase. I do see that temperature data is overlain on the plot of SSB in Figure 1, however, this is by no means sufficient to make generalizations on the influence of temperature on SSB. This

could be more formally addressed using temperature or climate indices as environmental data within population models.

- The critique I made in the previous bullet point leads me to think about the title of this manuscript. Are the authors truly disentangling a puzzle in population dynamics? Growth models are certainly a central function in population dynamics models, but I think the use of this term in the title leads the reader to believe there will be more to the analysis than simply investigating how temperature influences larval and juvenile growth in different populations of sardine across the North Pacific Ocean. A more comprehensive integration of this new information on growth and metabolism into population models is suggested with the current title, and since that is lacking I suggest re-focusing the title on the content that is covered in the manuscript: temperature and climate effects on growth and metabolic proxies.

- [Line 232] Again, size-at-age, growth, and population biomass increases are all very different, and I'm not sure how this manuscript can make definitive conclusions on population-level increases in biomass or abundance based on temperature effects on size-at-age, growth, and metabolism. Usually that requires a population model or stock assessment model.

-Response: The reviewer makes an important point. While this work has implications for the responses of populations to changing environmental conditions through by influencing the survival of early life-history stages, we have not made an effort here to explore the population responses using a numerical model. However, we have two reasons to believe that our conclusion is not a stretch. First, it has been shown in a number of studies that growth and survival during early life stages are strongly connected (Anderson, 1988; Sogard, 1997). Individuals found from the stomachs of predators often show slower growth than conspecifics collected in open waters in the same region (Takasuka et al., 2003). CA sardine recruits that survived through the mass mortality period had faster growth compared to those collected during larval and early juvenile stages (Takahashi et al., 2008a). Somatic growth rates around the early juvenile stage were strongly correlated with recruitment of JP sardine during 1996-2003 ($R^2 > 0.8$; Takahashi et al., 2008b). Although the strength of the correlation temporally varied when analysed for longer periods (Furuichi et al., 2020), the positive effect was consistently evident. These lines of evidence suggest that fast growths are essential to improve early life survival and recruitment, which was the reason we analysed the response of growth. To strengthen this point of view, we applied additional analysis of sea surface temperatures around the main habitat of JP and CA sardines during spring to summer and their log recruitment residual (LNRR), the early life survival index that takes density-dependent effects into account (Supplementary Figure 8). The sea surface temperatures showed significantly negative and positive, respectively, correlation with LNRR of JP and CA sardine. This is consistent with a number of studies that repeatedly and rigorously tested relationships between temperature and sardine production (e.g., Noto and Yasuda, 1999; Yatsu et al., 2005; Sugihara et al., 2012; Deyle et al., 2014; Lindgren et al., 2013; Nakayama et al., 2018). These suggest that cooler and

warmer habitat temperature during 4 months from hatch that increase growth rates of JP and CA sardines, respectively, improve survival and recruitment and lead to population growth. To stress these points, we added description of relationships between growth and survival and between habitat temperature and sardine productions to Introduction (Line 75-85), and results of analysis between SST and LNRR to the Results and Discussion (Line 241-247). Nevertheless, although these relationships can address the mechanisms driving different population responses to changing environmental conditions, the reviewer is correct to note that these are only components of the information necessary to assess the population dynamics, and a quantitative assessment of the population dynamics is not something undertaken here. Hence, we have tried to relax the statements concerning population dynamics in the three locations the reviewer has identified above and instead focus on the growth and metabolism that are more specifically considered in our work (Line 34-35, 251-252, 337-341).

- [Lines 64-67] Can the authors more transparently distinguish western and eastern coasts vs. boundary currents? I understand what they mean, but a reader may associate the eastern boundary current with the south/east coast, whereas I believe it should correspond to the western coast. A simple parenthetical note relating coasts and associated currents should suffice.

-Response: The sentence was modified to “A recent study of sardine off South Africa that includes both a western and an eastern boundary current system (Fig. 1b) showed that sardine off the south and east coasts (the warm, western boundary current system) grew faster under approximately 3 °C warmer water than those off the west coast (the cool, eastern boundary current system)” as suggested (Line 92-96).

- [Lines 75-76] Please specify inner ear or vestibular system.

-Response: The sentence was modified to “Application of novel analyses on otoliths, the calcium carbonate structures in the fish’s inner ear, ...” as suggested.

- [Line 79] Citing the author’s name and including the superscript citation number seems redundant (“Hoie et al.¹⁶”, especially when next to a citation that does not include this (17).

-Response: We appreciate the reviewer for checking the manuscript in detail. We deleted the author’s name.

- [Line 114] Why use a parenthetical citation here (19) instead of superscript?

-Response: We appreciate the reviewer for checking the manuscript in detail. Modified to superscripts.

- [Lines 115-117] Can the authors report if the South African samples only consisted of individuals that lived to 100 dph, or if this is just the timespan that growth histories matched JP and CA samples? If the latter, it is probably a good idea to plot all of the data (>100 dph) so the reader can see how growth histories deviate from JP and CA samples after 100 dph .

-Response: The South African samples are all adults that lived longer than 100 dph and probably longer than 150 dph. Although it is ideal to show longer data for these individuals, we could not because the otolith increment width measurements until 150dph were performed only for very limited number of samples (especially for the west coast samples: only 4 out of 51) and thus those data have a precision that is incongruent with the rest of the records. This is because the main focus of the SA sardine study (Sakamoto et al., 2020) was to discover differences in larval and early juvenile stages that are more likely to reflect environments of their origins, and it was only data for < 100 dph that were published in Sakamoto et al. (2020). Thus, we did not try to show the timespan that growth histories matched JP and CA samples but simply consider better to show only reliable data (< 100 dph).

- I think it needs to be stressed somewhere in the main text that Moto is a proxy, not a measurement, as it is being derived from previously published data on d13Cdiet and d13CDIC, not measured through samples collected by the authors in parallel to otolith sampling. Therefore, inferences made corresponding to Moto are based on assumptions and post-hoc treatment of previously published data, and may not exactly reflect reality.

-Response: We added description to Introduction that we used previously published data on d13Cdiet and d13CDIC to estimate M_{oto} to stress this point as suggested (Line 133-137).

- [Line 152] ANCOVA requires linearity of the regressions, and the polynomial regression relationships of 95th & 5th percentiles of Moto plotted in Figure 3a, 3d, and 3g do not all look linear. It's also hard to decipher linearity vs. nonlinearity in the Extended Data Figure 5 of the Appendix. If the authors have violated ANCOVA's assumption of linearity, they may need to find an alternative test or method to use (e.g., maybe AIC model selection exercise?). Or perhaps the regression of all data approaches linearity. If so, please plot the linear regression for each area as a dotted line colored according to area.

-Response: To address reviewer's concerns and follow reviewer's suggestions, we changed the analysis to use GLM instead of ANCOVA, because GLM has flexibility to deal with different data distributions by using their function link and evaluate the models by AIC. Although the nonlinearity of temperature-dependent trend is actually observed from the 95th and 5th percentile of data as the reviewer suggested, we evaluate the linearity of M_{oto} and temperature based on all data points. Gaussian function family function was used as overall M_{oto} distribution showed no significant discrepancy from Gaussian distribution (Shapiro-Wilk test, $w = 1.00$, $p > 0.05$). Theoretically, the

relationship between metabolism and temperature tends to show a linear trend after the metabolic rate is log-transformed. Thus, we applied “identity (data without transformed)” and “log (data transformed)” links to evaluate if model shows a better linearity with data transformation. Based on AIC, however, the result showed Moto have a better linearity without data transformation (Supplementary Table 7). We, therefore, used “identity” links for the further model selection. Model selection based on AIC was performed for models including temperature, region (JP and CA sardines), life history stages (larvae, early juvenile and late juvenile) and interactions of these factors. The full model including all the interactions had the lowest AIC (Supplementary Table 7). As the diagnostic for the full model showed normality and homogeneity of residuals (Supplementary Fig. 5), we selected this model for interpretation. The CA sardine at the larval stage as the baseline, we found only JP sardine at early and late juvenile stages has relatively higher M_{oto} values, and the temperature-dependent slope is significant gentler in JP sardine at early and late juvenile stages (Supplementary Table 8). We added detailed description for these analyses in Method (Line 664-687) and modified the description in the Results and Discussion (Line 190-194).

- [Line 163, Figure 3] Moto requires a subscript on oto. Also, ‘Juvenile I’ is not a useful moniker and kind of clunky after “Early Juvenile” – perhaps “Late Juvenile” depending on what the authors are trying to convey? In (a), (d), and (g), please make the JP lines red instead of black to match the rest of the figure.

-Response: We replaced “Juvenile I” with “Late juvenile” across the manuscript as suggested. Life history stage names used in this study were defined in Smith (1992), although it seems they are not used commonly. The lines in Fig. 3 were changed to brighter red to avoid potential misinterpretations.

- [Line 184, Figure 4] The sentence starting ”Significant regression lines..” in the caption is incomplete.

-Response: We appreciate the close checking of the manuscript. We completed the sentence as “Significant regression lines ($p < 0.05$) and 95% confidence intervals are shown as solid lines and shades.”.

- [Lines 209-219] The authors discuss how the optimal temperatures (green bars) correspond to the individuals with the largest sizes in each JP case (Fig 4a,c,e), but certainly not in the CA case. This sends a mixed message that optimal temperature may not be a universally important determinant of optimal growth. Alternative to the authors’ discussion on the additional importance of prey availability, this disconnection could be related to the assumptions made in generation of Moto, which would suggest that the estimates of Moto may not be the best for the CA region. This discussion on the discontinuities in main findings between study regions should be in the main text.

-Response: We appreciate for this important suggestion and explicitly referred to the possibility that

the use of literature values in M_{oto} generations may have affected the result for CA sardine as suggested (Line 287-289). However, we want to clarify: (1) As the discrepancy between optimal temperature and optimal growth is widely observed in literature (e.g. Righton et al., 2010; Audzijonyte et al., 2020), it does not necessarily indicate the flaws of the method but may rather suggest mechanical differences in physiological response. The underlying mechanism of the discrepancy has not been well investigated primary because of the difficulty to measure exact energy use for marine ectotherms in the field. We, therefore, tried to describe this by using M_{oto} as proxy to investigate the optimal temperature for sardines based on the M_{oto} range (temperature with highest metabolic diversity) and use otolith increment analyses to investigate the growth. (2) We can clearly explain how different responses were formed. In the case of optimal growth at optimal temperature (JP population), it means that total energy use of fish is reaching the maximum energy ceiling (maximum metabolic rate). The energy expenditure for growth is constrained by the width of aerobic scope and thus growth slows down in temperatures above optimal temperature. In the case of higher growth rates with temperature increases even over the optimal temperature (CA population), it means fish reserves available scope for energy expenditure across the temperature range. As the energy expenditure for growth would not be primary limited by the width of aerobic scope in this case, it is possible to increase energy expenditure for growth in temperatures above optimal temperature. According to estimates of M_{oto} values between JP and CA population, we found relatively higher values in JP than ones in CA populations even after taking the effect of temperature into account, further supporting that JP population tend to reach the energy ceiling but CA does not and have aerobic scope to increase energy expenditure.

Considering the reviewer's opinion "the estimates of M_{oto} may not be the best for the CA region", we reply in two aspects. First, is the estimate of M_{oto} accurate and precise? The value is calculated based on $d^{13}C$ values of DIC and prey, for which we considered different conditions for both populations using literatures from each region. After evaluating the uncertainty of the M_{oto} generation, we still have the same pattern and consequence, indicating the reliability of M_{oto} values. Second, is M_{oto} as reliable proxy to investigate the optimal temperature? We do see a clear pattern of metabolic diversity varying across the range of temperature. Also, the obtained optimal temperature corresponds to the temperature of their distribution range along the ontogenetic stages. Therefore, the method and estimate are applicable to our target populations. Although the change of growth in the CA population does not reach a peak at optimal temperature, it does not mean M_{oto} estimates are unsuitable for CA population but indicate mechanical differences in physiological response compared to the JP population. We have added more explanations in Results and Discussion to emphasise these points (Line 258-296).

- [Line 210] Is Figure 4 actually showing slower or faster growth relative to optimal growing temperature, or smaller and larger sizes relative to this? I think this is an important distinction that needs to be made throughout the manuscript. Otolith increment width conveys growth rate, as does the slope (or growth coefficient parameter) of a size-at-age relationship. However, the relationship between standard length and temperature is plotted, and there is no indication of a change in slope in this relationship at temperature below or above the optimal (green box), I believe the sentence I reference here is talking about size, not growth (i.e., “largest sizes of CA sardine were consistently larger at temperatures above the optimal temperature”). The authors may want to adjust the conclusions on growth made in this paragraph to account for this distinction.

-Response: We thank the reviewer for raising this point. Figure 4 shows relationships between mean temperature and size, which is the integration of temperature and growth rate in each stage, respectively, exhibiting the accumulated effect of temperature on growth from hatch to each life stage. By following the ontogenetic changes of temperature-size relationship and the shifts of optimal temperature, it is possible to infer the relationship between growth rate in each stage (i.e., slope) and optimal temperature. As questioned by the reviewer, however, this is slightly an indirect way to show the temperature-growth (slope) relationship. We, therefore, added a supplementary figure that shows the direct comparison between temperature-growth rate relationship and optimal temperature for each life stage (Supplementary Figure 7). Almost identical conclusions can be derived. Growth rates of JP peaked around temperatures close to the optimal temperatures throughout larval and juvenile stages, and growth rates of CA were higher in temperatures above the optimal temperatures, perhaps except for later juvenile stage in which the temperature-growth rate relationship was unclear (Supplementary Figure 7). We added description for this result (Line 224-228). Thus, we believe the descriptions relating optimal temperature and growth rates here are valid. Nevertheless, we consider it better to show relationships of size and mean temperature in the main text to clearly express our message; while temperature-growth rate relationship differs among life stages, mean growth rate throughout larval and juvenile stages is faster under cooler conditions for JP and under warmer conditions for CA sardine.

- [Line 227] ‘change’ should be ‘changing’ I believe.

-Response: The reviewer is correct. Modified.

Reviewer #3 (Remarks to the Author):

The authors produced quite an interesting study to compare two populations of sardines based on growth increment width, and stable isotope analysis of otoliths that could reveal the level of metabolism © and ambient temperatures (O). However, the authors did not study the survival of larvae

at different temperatures. Hence their conclusions about determinants of population growth of sardines (in terms of biomass) are not correct. Based on their data, the authors could definitely speak about differences in individual growth rates, but not in population growth. Then I wonder if they could achieve their goal to 'disentangle the puzzle' of biomass variability of sardines at both sides of the North Pacific.

I would suggest that the authors need to bring the evidence of survival variability, or downgrade their goal to study individual growth of both populations. In both cases the ms requires major revision.

-Response: We thank the reviewer for important criticisms that helped us to improve our manuscript. The concern for the link between individual growth rates and population sizes was also a point emphasized by the second reviewer. Both Reviewers 2 and 3 have made valid points about the caveats necessary in relating individual growth, metabolism, and survival to population-level dynamics. As noted in response to the second reviewer, changes were made to the text to focus on the consequences for metabolism and growth rather than population size.

Specific comments:

Lines 32-34. I don't think that you have studied survival of larvae and small juveniles here relative to temperatures? In this case how can you judge the fluctuations of biomass based just on individual growth rates?

(moved from later parts as these are well related) Lines 191-196. How do you correlate the individual growth of fish (somatic growth) with the growth of population (which is somatic growth + survival rate)? I don't think that you presented here any data on increased survival of either population of sardines at different temperatures?

Lines 228-232. I don't think that you did a correct conclusion here. Even if your findings are correct, that means that sardines of both populations grow differently, but it does not explain their population growth! You need to discuss that.

-Response: The reviewer makes an important point. While this work has implications for the responses of populations to changing environmental conditions through by influencing the survival of early life-history stages, we have not made an effort here to explore the population responses using a numerical model. However, we have two reasons to believe that our conclusion is not a stretch. First, it has been shown in a number of studies that growth and survival during early life stages are strongly connected (Anderson, 1988; Sogard, 1997). Individuals found from the stomachs of predators often show slower growth than conspecifics collected in open waters in the same region (Takasuka et al., 2003). CA sardine recruits that survived through the mass mortality period had faster growth compared to those

collected during larval and early juvenile stages (Takahashi et al., 2008a). Somatic growth rates around the early juvenile stage were strongly correlated with recruitment of JP sardine during 1996-2003 ($R^2 > 0.8$; Takahashi et al., 2008b). Although the strength of the correlation temporally varied when analysed for longer periods (Furuichi et al., 2020), the positive effect was consistently evident. These lines of evidence suggest that fast growths are essential to improve early life survival and recruitment, which was the reason we analysed the response of growth. To strengthen this point of view, we applied additional analysis of sea surface temperatures around the main habitat of JP and CA sardines during spring to summer and their log recruitment residual (LNRR), the early life survival index that takes density-dependent effects into account (Supplementary Figure 8). The sea surface temperatures showed significantly negative and positive, respectively, correlation with LNRR of JP and CA sardine. This is consistent with a number of studies that repeatedly and rigorously tested relationships between temperature and sardine production (e.g., Noto and Yasuda, 1999; Yatsu et al., 2005; Sugihara et al., 2012; Deyle et al., 2014; Lindgren et al., 2013; Nakayama et al., 2018). These suggest that cooler and warmer habitat temperature during 4 months from hatch that increase growth rates of JP and CA sardines, respectively, improve survival and recruitment and lead to population growth. To stress these points, we added description of relationships between growth and survival and between habitat temperature and sardine productions to Introduction (Line 75-85), and results of analysis between SST and LNRR to the Results and Discussion (Line 241-247). Nevertheless, although these relationships can address the mechanisms driving different population responses to changing environmental conditions, the reviewer is correct to note that these are only components of the information necessary to assess the population dynamics, and a quantitative assessment of the population dynamics is not something undertaken here. Hence, we have tried to relax the statements concerning population dynamics in the three locations the reviewer has identified above and instead focus on the growth and metabolism that are more specifically considered in our work (Line 34-35, 251-252, 337-341).

Lines 36-95. The introductory part of the manuscript should be re-organised, as not all readers are familiar with sardines and their fluctuations in abundance. Start first with description of sardines and their life cycles (lines 91-93), and put much more detail into it – are Japanese and Californian sardines the same species? If yes, what are the connections between populations? What kind of abundance variability is characteristic to each population? And then, how you suggest to solve it – by looking at metabolism signatures in their otoliths.

-Response: We appreciate this suggestion and re-organised the Introduction to help readers not familiar to sardines following the reviewer's instruction. We now start with description of sardine distributions and characteristics of their biomass fluctuations (Line 38-52), then explain ecology of JP and CA sardines and relationship between temperature and biomass before showing the way of solution (Line

55-68). JP and CA sardines are different subspecies (*Sardinops sagax melanostictus* and *Sardinops sagax sagax*). While they may have connections with other subpopulations in western North Pacific (the Tsushima Warm Current subpopulation) and eastern North Pacific (the southern subpopulation), there are likely no connection between JP and CA sardines because of the vast distance between the habitats (Fig. 1a).

Lines 97-100. Please describe the sampling scheme in a bit more detail. You collected sardines from JP and CA in different years – why? How can you compare them in you collected them in different years, especially those aged 0+ and 1? I would like to see justification for that. I guess if you the goal was to sample populations only during biomass increases – you need to have the data during biomass decrease to compare?

-Response: We thank the reviewer for raising these points. Our aim was to use ecologically comparable samples. As there are no direct interactions or mixings between JP and CA sardines, it is more important to compare samples in similar population state and life-history stages to avoid inclusion of potentially relevant bias, for instance population density-dependent effect or survivor's bias on growth, rather than choosing samples in the same year and age. Thus, the samples of similar body length (10-15 cm SL for JP and 10-16 cm SL for CA) and life history stages (immature recruits) collected in periods including population growth, were used. Furthermore, as we analysed life history traits during the consistent life history stages (larval and most of the juvenile stages), comparison of the results is valid in ecological context. It would have been ideal to collect samples during biomass decrease and add more contrasts, although this is practically difficult because we need collections of more than 30 years to cover one sardine biomass fluctuation cycle. Nevertheless, samples from periods of biomass increase period likely cover year-classes that experienced favourable conditions for population-level growth and survival. This allows investigation of the effects of environmental variabilities on life history changes, which may be relevant to population growth. We added these descriptions in the last paragraph of Introduction (Line 121-141).

You have to also specify how did you estimate the age of sardines – by counting daily growth increments? Are they validated?

-Response: Ages were judged by their standard length (Line 362-363, 378-379). The age of JP sardine was later confirmed by the counts of daily increments to the otolith edge (Line 362-363).

Lines 101-102. You have to proof that the increment width in the otolith corresponds well with growth rates of the whole fish. Sometimes it is a good correlation, but sometimes it is not as the otolith growth is not directly correlated with amount of food consumed and transformed into body weight.

Line 107 – as above – you have to show that you have a good estimate of 'somatic growth' using

otolith growth increment widths.

-Response: To address the reviewer's concerns, we added explanations for these in text with key references that described daily periodicity of growth increments formation of sardines based on rearing experiments and close relationships between otolith radius and body length (CA sardine: Butler, 1987; JP sardine: Hayashi et al., 1989; Funaki and Nakategawa 2007, Line 105-108). Butler (1987) showed that the correlation between otolith radius and body length are stronger than that between otolith radius and body weight for CA sardine < 19cm SL, suggesting that otolith increment widths have strong correlation with somatic growth rather than with increase of weight for sardine samples used in this study (< 16cm SL).

Line 113. What are 'otolith radii'?

-Response: Although this might not be seen often, we refer to multiple "otolith radius"s as "otolith radii" because the plural form of "radius" is "radii".

Lines 115-117. What did you mean by 'analogous', and why it is a surprise to you?

-Response: We appreciate the reviewer for identifying the ambiguity of the text. To avoid the use of "analogous" and "surprising", we changed this sentence to "Despite the vast geographic distances that separate sardine populations in South Africa from those in the North Pacific, growth histories of otolith radii of JP sardine and sardine from the south-east coast of South Africa were similar until 100dph." (Line 150-153). We were surprised because otolith growth trajectories were similar despite the fact that the sardines inhabit completely different regions.

Line 126. Even if JP sardines live in the warmer environment and hence have higher metabolic rates, it does not necessarily mean that they have higher growth rates. You need to prove it or to present the evidence here (like optimum temperatures for juvenile growth).

-Response: The conclusion that JP sardine has higher growth rates is derived from the result that JP sardine had larger otolith radius (Fig. 2a), and hence grew faster during early life stages, not from the higher metabolic proxy M_{oto} .

Lines 180-182. I did not understand correctly what did you mean by 'mean growth from hatch to the end of juvenile stage'. How did you calculate that? Dividing mean length at the end of the juvenile stage by the DPH?

-Response: As body length is the integration of daily somatic growth, the "mean growth from hatch to the end of juvenile stage" is the back-calculated length divided dph at the end of the juvenile stage as the reviewer wrote. To make this point clearer, we inserted the sentence "Body length is the integration of daily somatic growth." (Line 219-220). We also added the results of comparison between growth

rate and temperature in each stage (Supplementary Figure 7), from which similar conclusion can be derived (Line 224-228).

Literature cited:

A. Audzijonyte, S. A. Richards, R. D. Stuart-Smith, G. Pecl, G. J. Edgar, N. S. Barrett, N. Payne, J. L. Blanchard, Fish body sizes change with temperature but not all species shrink with warming. *Nat. Ecol. Evol.* **4**, 809-814 (2020).

J. L. Butler, Comparison of larval and juvenile growth and larval mortality rates of Pacific sardine and northern anchovy and implications for species interactions. *PhD Thesis, University of California*. 242p. (1987).

E. R. Deyle, M. Fogarty, C. H. Hsieh, L. Kaufman, A. D. MacCall, S. B. Munch, C. T. Perretti, H. Ye, G. Sugihara, Predicting climate effects on Pacific sardine. *Proc. Natl. Acad. Sci. U. S. A.* **110**, 6430-6435 (2013).

O. Funaki, H. Nakategawa, Daily growth increments in otoliths of Japanese sardine. *Kanagawa Prefectural Fisheries Technology Center Reports* **2**, 97-100 (2007) (in Japanese).

S. Furuichi, Y. Niino, Y. Kamimura, R. Yukami, Time-varying relationships between early growth rate and recruitment in Japanese sardine. *Fisheries Research*. **232**, 105723 (2020).

A. Hayashi, Y. Yamashita, K. Kawaguchi, T. Ishii, Rearing method and daily otolith ring of Japanese sardine larvae. *Nippon Suisan Gakkaishi* **55**, 997-1000 (1989).

M. Lindegren, D. M. Checkley Jr, T. Rouyer, A. D. MacCall, N. C. Stenseth, Climate, fishing, and fluctuations of sardine and anchovy in the California Current. *Proc. Natl. Acad. Sci. U. S. A.* **110**, 13672-13677 (2013).

S. Nakayama, A. Takasuka, M. Ichinokawa, H. Okamura, Climate change and interspecific interactions drive species alternations between anchovy and sardine in the western North Pacific: Detection of causality by convergent cross mapping. *Fish. Oceanogr.* **27**, 312-322 (2018).

M. Noto, I. Yasuda, Population decline of the Japanese sardine, *Sardinops melanostictus*, in relation to sea surface temperature in the Kuroshio Extension. *Can. J. Fish. Aquat. Sci.* **56**, 973-983 (1999).

D. A. Righton, K. H. Andersen, F. Neat, V. Thorsteinsson, P. Steingrund, H. Svedäng, K. Michalsen,

- H. Hinrichsen, V. Bendall, S. Neuenfeldt, Thermal niche of Atlantic cod *Gadus morhua*: limits, tolerance and optima. *Mar. Ecol. Prog. Ser.* **420**, 1-13 (2010).
- T. Sakamoto, C. D. van der Lingen, K. Shirai, T. Ishimura, Y. Geja, J. Peterson, K. Komatsu, Otolith $\delta^{18}\text{O}$ and microstructure analyses provide further evidence of population structure in sardine *Sardinops sagax* around South Africa. *ICES J. Mar. Sci.* **77**, 2669-2680 (2020).
- P. E. Smith, Life-stage duration and survival parameters as related to interdecadal population variability in Pacific sardine. *CalCOFI Rep.* **33**, 41-47 (1992).
- S. M. Sogard, Size-selective mortality in the juvenile stage of teleost fishes: a review. *Bull. Mar. Sci.* **60**, 1129-1157 (1997).
- G. Sugihara, R. May, H. Ye, C. H. Hsieh, E. Deyle, M. Fogarty, S. Munch, Detecting causality in complex ecosystems. *Science.* **338**, 496-500 (2012).
- M. Takahashi, H. Nishida, A. Yatsu, Y. Watanabe, Year-class strength and growth rates after metamorphosis of Japanese sardine (*Sardinops melanostictus*) in the western North Pacific Ocean during 1996–2003. *Can. J. Fish. Aquat. Sci.* **65**, 1425-1434 (2008).
- A. Takasuka, I. Aoki, I. Mitani, Evidence of growth-selective predation on larval Japanese anchovy *Engraulis japonicus* in Sagami Bay. *Mar. Ecol. Prog. Ser.* **252**, 223-238 (2003).
- A. Takasuka, Y. Oozeki, I. Aoki, R. Kimura, H. Kubota, H. Sugisaki, T. Akamine, Growth effect on the otolith and somatic size relationship in Japanese anchovy and sardine larvae. *Fisheries Science.* **74**, 308-313 (2008).
- A. Yatsu, T. Watanabe, M. Ishida, H. Sugisaki, L. D. Jacobson, Environmental effects on recruitment and productivity of Japanese sardine *Sardinops melanostictus* and chub mackerel *Scomber japonicus* with recommendations for management. *Fish. Oceanogr.* **14**, 263-278 (2005).

REVIEWER COMMENTS

Reviewer #1 (Remarks to the Author):

The authors have carefully addressed the reviewer comments in this revision, and given comprehensive justifications for any changes implemented in response to the feedback. Justifications for "no change" are also comprehensive. In most cases these are reasonable arguments, but there are a few aspects that should be considered more carefully:

1) remove the mention of prey conditions from the abstract. although literature values are used for Moto and this implies knowledge of prey conditions (although these literature values were not adjusted for different years to coincide with the samples or projections) - prey was not directly a focus in this study and mentioning it in the abstract is somewhat misleading
2) the use of backcalculation from annual increments is rarely validated directly - and more so for primary increments. With a missing anchor point for initial size, you still have the same problem of decoupling of otolith and somatic growth - thus using width as a proxy for growth (or better, using the variation from mean width - as in chronology studies - a BLUP) involves fewer assumptions
3) the integration of the SA comparison is still weak, and there is still no indication in the abstract that this more general E/W phenomenon is being tested. At the first mention (line 95) there needs to be some indication that this may be used to test the wider implications of the JP-CA comparison. At lines 167-171 the direct comparison is presented, but there is some ambiguity in the wording at the end: "These are consistent with the differences between populations of the western and eastern boundary current systems that support sardine near South Africa, likely reflecting the differing ocean environments in the two systems." Does the "two systems" refer to Pacific/SA or to E/W? please specify

At lines 257-260 is the stronger statement about the potential that E/W difference in response is more general and global. This in fact should be part of the problem statement brought in much earlier in the manuscript. If a general pattern is being observed, then it makes sense to test it in other systems. That rationale is still missing, or else its statement is so diffused throughout the text that the reader misses the (important) point

For example: at Line 418 should add in "more generally" to emphasize

4) justification for the link between early growth and population responses may be evaluated by other reviewers - this may be true for sardines and other small pelagics (in warmer waters), but is not the case for longer-lived species where y1 growth and survival are more linked to recruitment and population dynamics

Reviewer #2 (Remarks to the Author):

I've already reviewed this, so my comments will be brief. The authors addressed my critiques adequately. I have no further issues with the manuscript.

Reviewer #3 was unavailable to re-review so reviewer #1 was asked to comment on your responses to reviewer #3:

I've read through the revised MS again, along with the rebuttal letter and I believe that the authors have taken into account many of the comments raised by all of us reviewers - to some extent. The rebuttal is typical of responses that present the equivalent of an entire discussion section within their rebuttal, rather than spending that time rewriting their manuscript to take account of the comments.

After reading the new version of the manuscript, I felt that the authors still had not justified:

1) The inclusion of the SA populations in the main body - this is nothing more than a diversion here and is more fitting to something mentioned in passing in the discussion: "as also likely in SA sardines..." They could have picked up any sardines from other systems and compared growth trajectories

2) The growth comparisons are still not convincing as a series of ANOVAs - because they don't

take into account repeated measures – unless they state explicitly how the size at age (at each subsequent age bin) is not a repeated measure. They would have to try to sample randomly from the individuals in each bin. Otherwise there are only 2 groups (JP and CA) and the individual fish within each group are not replicates

3) The whole point of reviewer 3 – the linking to population dynamics – has been answered with some toning down of the conclusions, but also with a lengthy justification attempt to show that larval and juvenile growth is linked to survival and recruitment. That's fine, but they are also arguing in the ms that west and east sardines exhibit different growth patterns driven by temperature, and therefore would expect (by their argument) that recruitment would differ as well. I believe that the authors are ok to talk about recruitment (or more properly, survival to juvenile stage) – but not population biomass or population growth. The difference is that most of the papers they cite in the ms and in the rebuttal limit the link to being between growth rate and survival through to metamorphosis. There is also a reliance on the studies that show that the individuals surviving after mass mortality events are those that grew faster when they were younger. I think the authors are safe to talk about survival following faster growth, which often follows higher metabolism (but not always), but really still need to tone down the further link to population dynamics both in the introduction and in the discussion. After all, they do not have any data showing the proportion of the samples that exhibit the faster or slower growth/metabolism

Reply to the reviewer comments

We are truly grateful to the reviewers for their beneficial comments and suggestions on our manuscript. The comments have helped us to improve the paper considerably. Our replies to the comments are listed below in due order with red font.

REVIEWER COMMENTS

Reviewer #1 (Remarks to the Author):

The authors have carefully addressed the reviewer comments in this revision, and given comprehensive justifications for any changes implemented in response to the feedback. Justifications for "no change" are also comprehensive. In most cases these are reasonable arguments, but there are a few aspects that should be considered more carefully:

-We are grateful to the reviewer for identifying the aspects that are still not well justified and for providing another review on behalf of Reviewer #3.

1) remove the mention of prey conditions from the abstract. although literature values are used for Moto and this implies knowledge of prey conditions (although these literature values were not adjusted for different years to coincide with the samples or projections) - prey was not directly a focus in this study and mentioning it in the abstract is somewhat misleading

-We removed the mention of prey conditions from the abstract as suggested. We mentioned the comparison with SA sardines instead.

2) the use of backcalculation from annual increments is rarely validated directly - and more so for primary increments. With a missing anchor point for initial size, you still have the same problem of decoupling of otolith and somatic growth - thus using width as a proxy for growth (or better, using the variation from mean width - as in chronology studies - a BLUP) involves fewer assumptions

-We agree with the reviewer that using otolith increment width involves fewer assumptions. Therefore, we provide an additional analysis comparing increment width and temperature at each stage (Supplementary Fig. 12) which has similar indications to the comparison between estimated somatic growth and temperature (Supplementary Fig. 10). Nevertheless, we believe it is better to retain the comparison of back-calculated length and temperature in the main text because of the robustness of back-calculation and the known bias for the otolith: size relationship in Japanese sardine (larger otoliths in slower-growing individuals, Takasuka et al., 2008). Based on rearing experiments of field collected eggs, Lasker (1964) showed the SL of CA sardine at 6–8 dph ranged between 3.8 to 6.5 mm, and

Matsuoka and Mitani (1989) showed the total length at 2–4 dph ranged between 4.8 to 6.2 mm, corresponding to 4.7 to 6.1 mm in SL. Therefore, we conducted Monte Carlo simulations (10,000 times) to estimate the uncertainties of back-calculated SL, assuming that the first SLs fall between 3.8 to 6.5 mm for both sardines. Standard deviations of the temporal values were presented as the uncertainty of each SL_n estimation, which varied between 0.51 and 0.73 at the end of the larval stage (JP: 45 dph, CA: 60 dph), between 0.34 and 0.64 at the end of the early juvenile stage (JP: 75 dph, CA: 90 dph) and between 0.20 and 0.53 at the end of the late juvenile stage (JP: 105 dph, CA: 120 dph). Because these values are significantly small compared to the variabilities of estimated size among individuals (standard deviations: 4.2, 8.1 and 8.3 in JP and 5.5, 9.1 and 10.3 in CA for the end of the larval, early juvenile and late juvenile stages, respectively), we consider that the comparisons between the estimated size and temperature is robust against the uncertainties of back-calculated length. We added this consideration to the Methods (lines 434-447).

3) the integration of the SA comparison is still weak, and there is still no indication in the abstract that this more general E/W phenomenon is being tested.

(moved from the later part as it is strongly related)

1) The inclusion of the SA populations in the main body – this is nothing more than a diversion here and is more fitting to something mentioned in passing in the discussion: “as also likely in SA sardines....” They could have picked up any sardines from other systems and compared growth trajectories

-We appreciate the suggestions and criticisms regarding the integration of South African sardines to the narrative throughout the manuscript. To better incorporate the comparison following the suggestions, we inserted or modified sentences regarding the generality of east-west difference and the comparison with SA sardine in the Introduction and Discussion. (lines 35-37, 58-59, 106-109, 130-133, 157-160 and 343-346).

At the first mention (line 95) there needs to be some indication that this may be used to test the wider implications of the JP-CA comparison.

-To address this concern, we added the sentence “As the differences follow the distinct oceanographic conditions of the western and eastern boundary current systems, they may be the common features of populations in the subtropical boundary current systems around the globe.” (lines 106-109).

At lines 167-171 the direct comparison is presented, but there is some ambiguity in the wording at the end: "These are consistent with the differences between populations of the western and eastern boundary current systems that support sardine near South Africa, likely reflecting the differing ocean environments in the two systems." Does the "two systems" refer to Pacific/SA or to E/W? please specify

-It was E/W. We modified the sentence as "... , likely reflecting the differing ocean environments in the western (warmer) and eastern (cooler) boundary current systems." (lines 187-188).

At lines 257-260 is the stronger statement about the potential that E/W difference in response is more general and global. This in fact should be part of the problem statement brought in much earlier in the manuscript. If a general pattern is being observed, then it makes sense to test it in other systems. That rationale is still missing, or else its statement is so diffused throughout the text that the reader misses the (important) point

-We tried to clarify that the aim of this study is to understand the general E/W difference in population response by modifying early and late paragraphs in the Introduction (lines 58-59 and 131-134) and explicitly described that the results of JP-CA would be compared to SA results to seek for general implications (lines 157-160).

For example: at Line 418 should add in "more generally" to emphasize

-Added.

4) justification for the link between early growth and population responses may be evaluated by other reviewers - this may be true for sardines and other small pelagics (in warmer waters), but is not the case for longer-lived species where y1 growth and survival are more linked to recruitment and population dynamics

(moved from the later part as it is strongly related)

3) The whole point of reviewer 3 – the linking to population dynamics – has been answered with some toning down of the conclusions, but also with a lengthy justification attempt to show that larval and juvenile growth is linked to survival and recruitment. That's fine, but they are also arguing in the ms that west and east sardines exhibit different growth patterns driven by temperature, and therefore would expect (by their argument) that recruitment would differ as well. I believe that the authors are ok to talk about recruitment (or more properly, survival to juvenile stage) – but not population biomass or population growth. The difference is that most of the papers they cite in the ms and in the rebuttal limit the link to being between growth rate and survival through to metamorphosis. There is also a reliance on the studies that show that the individuals surviving after mass mortality events are those that grew faster when they were younger. I think the authors are safe to talk about survival following faster growth, which often follows higher metabolism (but not always), but really still need to tone down the further link to population dynamics both in the introduction and in the discussion. After all, they do not have any data showing the proportion of the samples that exhibit the faster or slower growth/metabolism

-We accept the criticism that populations are not regulated only by recruitment and other factors such as growth and survivals of the recruits are also important. To address this concern, we described in the Introduction that variability in the recruitment is one of several factors regulating fish populations, along with density-dependent mortality of the recruits (lines 83-84, 340-342 with a related reference provided). Following the suggestion that we are safe to discuss variabilities of recruitment but not population growth, we have modified the text to emphasize that growth-survival processes can have significant impact on recruitment and therefore “potentially” affect population fluctuation throughout the manuscript (lines 88-92, 272-274, 332-334).

I've read through the revised MS again, along with the rebuttal letter and I believe that the authors have taken into account many of the comments raised by all of us reviewers – to some extent. The rebuttal is typical of responses that present the equivalent of an entire discussion section within their rebuttal, rather than spending that time rewriting their manuscript to take account of the comments.

After reading the new version of the manuscript, I felt that the authors still had not justified:

1) The inclusion of the SA populations in the main body – this is nothing more than a diversion here and is more fitting to something mentioned in passing in the discussion: “as also likely in SA sardines....” They could have picked up any sardines from other systems and compared growth trajectories

-Moved and answered above.

2) The growth comparisons are still not convincing as a series of ANOVAs – because they don't take into account repeated measures – unless they state explicitly how the size at age (at each subsequent age bin) is not a repeated measure. They would have to try to sample randomly from the individuals in each bin. Otherwise there are only 2 groups (JP and CA) and the individual fish within each group are not replicates

-We agree that the effect of repeated measures were not taken into account. To address this issue, we changed the approach to mixed-modelling of the increment width data. The model showed that increment widths of JP and SA_south-east during 21-60 dph and CA and SA-west during 31-60 dph, respectively, were not significantly different, thereby creating the general similarity of OR growth during 0 to 100 dph. We added Supplementary Figures 5 and 6 and Supplementary Table 5 to show these results, and modified the Methods (lines 661-674). Furthermore, comparisons of M_{oto} and experienced temperature were also changed to the mixed-effects modelling approach as these are also repeatedly measured. We added Supplementary Figures 6 and 7 and Supplementary Tables 6 and 7 to show the results and modified the Methods (lines 688-701).

3) The whole point of reviewer 3 – the linking to population dynamics – has been answered with some toning down of the conclusions, but also with a lengthy justification attempt to show that larval and juvenile growth is linked to survival and recruitment. That's fine, but they are also arguing in the ms that west and east sardines exhibit different growth patterns driven by temperature, and therefore would expect (by their argument) that recruitment would differ as well. I believe that the authors are ok to talk about recruitment (or more properly, survival to juvenile stage) – but not population biomass or population growth. The difference is that most of the papers they cite in the ms and in the rebuttal limit the link to being between growth rate and survival through to metamorphosis. There is also a reliance on the studies that show that the individuals surviving after mass mortality events are those that grew faster when they were younger. I think the authors are safe to talk about survival following faster growth, which often follows higher metabolism (but not always), but really still need to tone down the further link to population dynamics both in the introduction and in the discussion. After all, they do not have any data showing the proportion of the samples that exhibit the faster or slower growth/metabolism

-Moved and answered above.

Reviewer #2 (Remarks to the Author):

I've already reviewed this, so my comments will be brief. The authors addressed my critiques adequately. I have no further issues with the manuscript.

-We appreciate the reviewer's continued efforts to examine the manuscript in detail and provide helpful comments to improve the description of our work and its implications.

References

R. Lasker, An experimental study of the effect of temperature on the incubation time, development, and growth of Pacific sardine embryos and larvae. *Copeia*. **2**, 399-405 (1964).

M. Matsuoka, T. Mitani, Hatching and rearing experiments on eggs of Japanese sardine, *Sardinops melanostictus*, collected from adjoining area of the Nagasaki harbor (Preliminary study). *Bull. Seikai Reg. Fish. Res. Lab.* **67**, 15-22 (1989). (in Japanese)

A. Takasuka, Y. Oozeki, I. Aoki, R. Kimura, H. Kubota, H. Sugisaki, T. Akamine, Growth effect on

the otolith and somatic size relationship in Japanese anchovy and sardine larvae. *Fisheries Science*. **74**, 308-313 (2008).

REVIEWERS' COMMENTS

Reviewer #1 (Remarks to the Author):

The authors have significantly improved the methodology and persuasiveness of the manuscript, and addressed all of the points raised in a sincere way. That's great because often the author/reviewer relationship devolves to a wholly argumentative one, without a more collegial "let's build a more convincing text here". I appreciate that. I really did like the E/W story and now it is much more convincing and set in context compared to the first version.

It's good to see the mixed modelling approach to comparisons to replace the ANOVAs

I still take issue with the back-calculations, but the inclusion of the width data and analysis is helpful to readers who would want to see direct responses. Though I'm not calling for any further changes, I would like to point out the contradiction in the rebuttal statement:

"Nevertheless, we believe it is better to retain the comparison of back-calculated length and temperature in the main text because of the robustness of back-calculation and the known bias for the otolith: size relationship in Japanese sardine (larger otoliths in slower-growing individuals, Takasuka et al., 2008)."

Since back-calculation relies on the otolith: size relationship, in fact it compounds the error compared to using increment widths directly. It is exactly those errors in back-calculations which led to the whole realisation of decoupling of otolith and somatic growth by Mosegaard and by Secor in the 1980s

Reply to the reviewer comments

We are truly grateful to the reviewer for the beneficial comments. Our replies to the comments are listed below in due order with red font.

REVIEWER COMMENTS

Reviewer #1 (Remarks to the Author):

The authors have significantly improved the methodology and persuasiveness of the manuscript, and addressed all of the points raised in a sincere way. That's great because often the author/reviewer relationship devolves to a wholly argumentative one, without a more collegial "let's build a more convincing text here". I appreciate that. I really did like the E/W story and now it is much more convincing and set in context compared to the first version.

It's good to see the mixed modelling approach to comparisons to replace the ANOVAs

I still take issue with the back-calculations, but the inclusion of the width data and analysis is helpful to readers who would want to see direct responses. Though I'm not calling for any further changes, I would like to point out the contradiction in the rebuttal statement:

"Nevertheless, we believe it is better to retain the comparison of back-calculated length and temperature in the main text because of the robustness of back-calculation and the known bias for the otolith:size relationship in Japanese sardine (larger otoliths in slower-growing individuals, Takasuka et al., 2008)."

Since back-calculation relies on the otolith:size relationship, in fact it compounds the error compared to using increment widths directly. It is exactly those errors in back-calculations which led to the whole realisation of decoupling of otolith and somatic growth by Mosegaard and by Secor in the 1980s

- We appreciate the reviewer's continued efforts to examine the manuscript in detail and provide helpful comments to improve the description of our work and its implications. We are also grateful for pointing out the issue in our rebuttal. We reviewed the works and discussion regarding potential errors in back-calculations and added the following sentences to the Methods section to address the issue (lines 439-447): "Nevertheless, the biological intercept method assumes a constant linear relationship between fish and otolith size within individual (Campana and Jones, 1992), which can vary depending on physiological or environmental conditions (Mosegaard et al., 1988; Secor and Dean, 1992). Therefore, to examine the relationships between temperature and growth, we used both otolith growth, which contains fewer assumptions, and back-calculated somatic growth as growth proxies. Since the

use of the two proxies did not show remarkable differences in the relationships between temperature and growth (Supplementary Figs. 11, 12), we mainly used the back-calculated SL in the discussion, which has a more direct ecological implication.”.

References

- S. E. Campana, C. M. Jones, Analysis of otolith microstructure data. *Otolith Microstructure Examination and Analysis. can Spec Publ Fish Aquat Sci.* **117**, 73-100 (1992).
- H. Mosegaard, H. Svedäng, K. Taberman, Uncoupling of Somatic and Otolith Growth Rates in Arctic Char (*Salvelinus alpinus*) as an Effect of Differences in Temperature Response. *Canadian Journal of Fisheries and Aquatic Sciences – Can. J. Fish. Aquat. Sci.* **45**, 1514-1524 (1988).
- D. H. Secor, J. M. Dean, Comparison of otolith-based back-calculation methods to determine individual growth histories of larval striped bass, *Morone saxatilis*. *Can. J. Fish. Aquat. Sci.* **49**, 1439-1454 (1992).